# Irrigation, damming, and streamflow fluctuations of the Yellow River

Zun Yin[1,2], Catherine Ottlé[1], Philippe Ciais[1], Feng Zhou[3], Xuhui Wang[3], Polcher Jan[2], Patrice Dumas[4], Shushi Peng[3], Laurent Li[2], Xudong Zhou[2,5,6], Yan Bo[3], Yi Xi[3], and Shilong Piao[4]

[1]Laboratoire des Sciences du Climat et de l'Environnement, IPSL, CNRS-CEA-UVSQ, Gif-sur-Yvette, France
[2]Laboratoire de Météorologie Dynamique, IPSL UPMC/CNRS, Paris 75005, France
[3]Sino-French Institute for Earth System Science, College of Urban and Environmental Sciences, Peking University, Beijing 100871, China
[4]Centre de Coopération Internationale en Recherche Agronomique pour le Développement, Avenue Agropolis, 34398 Montpellier CEDEX 5, France
[5]Institute of Industrial Science, The University of Tokyo, Tokyo, Japan
[6]State Key Laboratory of Hydrology-Water Resources and Hydraulic Engineering, Center for Global Change and Water Cycle, Hohai University, Nanjing 210098, China

**Correspondence:** Z. Yin
(zyin@princeton.edu)
Present affiliation: Geophysical Fluid Dynamics Laboratory, Princeton University, Princeton, New Jersey, USA

**Abstract.** The streamflows of the Yellow River (YR) is strongly affected by human activities like irrigation and dam operation. Many attribution studies focused on the long-term trends of streamflows, yet the contributions of these anthropogenic factors to streamflow fluctuations have not been well quantified with fully mechanistic models. This study aims to 1) demonstrate whether the mechanistic global land surface model ORCHIDEE is able to simulate the streamflows of this complex rivers with human activities using a generic parameterization for human activities, and 2) preliminarily quantify the roles of irrigation and dam operation in monthly streamflow fluctuations of the YR from 1982 to 2014 with a newly developed irrigation module, and an offline dam operation model. Validations with observed streamflows near the outlet of the YR demonstrated that model performances improved notably with incrementally considering irrigation (mean square error [MSE] decreased by 56.9%) and dam operation (MSE decreased by another 30.5%). Irrigation withdrawals were found to substantially reduce the river streamflows by approximately $242.8 \pm 27.8 \times 10^8$ m$^3$.yr$^{-1}$ in line with independent census data ($231.4 \pm 31.6 \times 10^8$ m$^3$.yr$^{-1}$). Dam operation does not change the mean streamflows in our model, but it impacts streamflow seasonality, more than the seasonal change of precipitation. By only considering generic operation schemes, our dam model is able to reproduce the water storage changes of the two large reservoirs LongYangXia and LiuJiaXia (correlation coefficient of ~0.9). Moreover, other commonly neglected factors, such as the large operation contribution from multiple medium/small reservoirs, the dominance of large irrigation districts for streamflows (e.g., the Hetao Plateau), and special management policies during extreme years, are

highlighted in this study. Related processes should be integrated in models to better project future YR water resources under climate change and optimize adaption strategies.

# 1 Introduction

More than 60% of all rivers in the world are disturbed by human activities (Grill et al., 2019) contributing altogether to approximately 63% of surface water withdrawal (Hanasaki et al., 2018). River water is used for agriculture, industry, drinking water supply, and electricity generation (Hanasaki et al., 2018; Wada et al., 2014), these usages being influenced by direct anthropogenic drivers and by climate change (Haddeland et al., 2014; Piao et al., 2007, 2010; Yin et al., 2020; Zhou et al., 2020). In order to meet the fast-growing water demand in populated areas and to control floods (Wada et al., 2014) reservoirs have been built up for regulating the temporal distribution of river water (Biemans et al., 2011; Hanasaki et al., 2006) leading to a massive perturbation of the seasonality and year-to-year variations of streamflows. In the mid-northern latitudes regions where a decrease of rainfall is observed historically and projected by climate models (Intergovernmental Panel on Climate Change, 2014), water scarcity will be further exacerbated by the growth of water demand (Hanasaki et al., 2013) and by the occurrence of more frequent extreme droughts (Seneviratne et al., 2014; Sherwood and Fu, 2014; Zscheischler et al., 2018). Thus, adapting river management is a crucial question for sustainable development, which requires comprehensive understanding of the impacts of human activities on river flow dynamics particularly in regions under high water stress (Liu et al., 2017; Wada et al., 2016).

The Yellow River (YR) is the second longest river in China. It flows across arid, semi-arid, and semi-humid regions, and the catchment contains intensive agricultural zones and has a population of 107 million inhabitants (Piao et al., 2010). With 2.6% of total water resources in China, the Yellow River Basin (YRB) irrigates 9.7% of the croplands (http://www.yrcc.gov.cn). Underground water resources are used in the YRB, but they only accounts for 10.3% of total water resources, outlining the importance of streamflow water for regional water use. A special feature of the YRB is the huge spatio-temporal variation of its water balance. Precipitation concentrates in the flooding season (from July to October) which concentrates ~60% of the annual discharge, whereas the dry season (from March to June) represents only ~10-20%. Numerous dams have been built up to regulate the streamflows intra- and inter-annually in order to control floods and alleviate water scarcity (Liu et al., 2015; Zhuo et al., 2019). The YRB streamflows are thus highly controlled by human water withdrawals and dam operations, making it difficult to separate the impacts of human and natural factors on the variability and trends.

Numerous studies documented the effects of anthropogenic factors on streamflows and water resources in the YRB by statistical approaches (e.g., Liu and Zhang (2002); Jin et al. (2017); Miao et al. (2011); Wang et al. (2006, 2018); Zhuo et al. (2019)). To further elucidate the mechanisms, physical-based land surface hydrology models including natural and anthropogenic factors are required. Many previous model studies only considered natural processes and YRB simulations were evaluated against naturalized streamflows (Liu et al., 2020; Xi et al., 2018; Yuan et al., 2018; Zhang and Yuan, 2020). YRB modeling studies simulating real streamflows and comparing their values to observed streamflows are scarce, the most important being from Jia et al. (2006); Tang et al. (2008). Yet, Jia et al. (2006) prescribed census irrigation and dam operation data as input of their model. Tang et al. (2008) included irrigation as a mechanism in their DBH model and investigated the long-term trends of streamflows, but they described the irrigation demand simply from satellite leaf area data, so that crop plant water requirements and phenology were not represented by physical laws. Several global hydrological models (GHMs) simulated both irrigation

and dam operation processes, and were applied for future projection of water resources regionally (Liu et al., 2019) or globally (Hanasaki et al., 2018; Wada et al., 2014, 2016). Those global GHM studies acknowledged the complex situation of the YRB where models' performances are limited, but none has focused on the sources of error or potential overlooked mechanisms in
this catchment.

To model present water resources in the YRB and make future projections, not only natural mechanisms, but also anthropogenic ones must be represented in a model. If a key mechanism is missing in a model, a calibration of its parameters to match observations can compensate for structural biases and projections may be erroneous. For example, the HBV model (Hydrologiska Byråns Vattenbalansavdelning) was well-calibrated with different approaches in 156 catchments in Austria but
failed in predicting streamflow changes due to climate warming (Duethmann et al., 2020). One of the key reasons being that the response of vegetation to climate change was missing in the model. In this study, we integrate two key anthropogenic processes (irrigation and dam operation) in the land surface model ORCHIDEE (ORganizing Carbon and Hydrology in Dynamic EcosystEms) which has a mechanistic description of plant-climate and soil water availability interactions and of river streamflows. Through a set of simulations with generic parameter values, we aim to preliminarily diagnose how irrigation and
dam operation improve the simulations of observed YRB streamflows. After making sure we understand the impact of adding these two new and crucial processes, the model will be calibrated against a suite of observations so that it can be applied for future projections.

Using a standard version of ORCHIDEE without irrigation nor dams, Xi et al. (2018) performed simulations with a 0.1° hypo-resolution atmospheric forcing over China (Chen et al., 2011). They attributed the trends of several river streamflows to
natural drivers from increased $CO_2$ and climate change and to land use change. Lacking irrigation and other human removals, their simulated results were higher than the observed streamflows for the YRB. By developing a crop module in ORCHIDEE (Wang et al., 2016; Wang, 2016; Wu et al., 2016), ORCHIDEE were able to provide precise estimation of crop physiology, phenology, and yield at both local and national scales as well as other site-based crop models (e.g., EPICs (Folberth et al., 2012; Izaurralde et al., 2006; Liu et al., 2007, 2016; Williams, 1995), CGMS-WOFOST (de Wit and van Diepen, 2008),
APSIM (Elliott et al., 2014; Keating et al., 2003), and DSSAT (Jones et al., 2003)) and land surface models (e.g., CLM-CROP (Drewniak et al., 2013), LPJ-GUESS (Smith et al., 2001; Lindeskog et al., 2013), LPJmL (Waha et al., 2012; Bondeau et al., 2007), and PEGASUS (Deryng et al., 2011, 2014) ) (Wang et al., 2017; Müller et al., 2017). ORCHIDEE-estimated irrigation accounts for potential ecological and hydrological impacts (e.g., physiological response of plants to climate change and short term drought episodes on soil hydrology) with respect to other land surface models (LSMs) and GHMs (Hanasaki et al., 2008;
Leng et al., 2015; Thiery et al., 2017; Nazemi and Wheater, 2015; Voisin et al., 2013). In a study focusing on China (Yin et al., 2020), ORCHIDEE was able to simulate irrigation withdrawals across China and to evaluate them against census data with a provincial-based spatial correlations of ~0.68. It successfully explained the decline of total water storage in the YRB. In this study, we add a simple module describing the dam operations to further improve the model over the YRB.

A simple dam operation model is developed and firstly coupled to ORCHIDEE to simulate the real streamflows in this study.
Similar to other GHMs and LSMs, our dam operation model is based on generic operation principles due to lack of related data. Recent dam models are developed from different perspectives, such as agent-based model River Wave (Humphries et al.,

2014), basin-specific model Colorado River Simulation System (Bureau of Reclamation, 2012), and the original dam module in the Variable Infiltration Capacity (VIC) model (Lohmann et al., 1998). However, the representation of dam operations in many global hydrological studies (e.g., Droppers et al. (2020); Haddeland et al. (2006, 2014); Hanasaki et al. (2018); Zhao et al.

(2016); Yassin et al. (2019); Wada et al. (2014, 2016)) are based on the ideas of Hanasaki et al. (2006). They categorized dams based on their regulation purposes (irrigation and non-irrigation). Irrigation-oriented rules adjust the dam retention to meet the irrigation demand downstream, while non-irrigation-oriented rules buffer floods and thus dampen the variability (Hanasaki et al., 2006). However, the water release target of a dam in the model of Hanasaki et al. (2006) is fixed at the beginning of the year and cannot adjust interactively to large intra- and inter-annual climate variations, which is a key feature of the YRB.

To overcome this limitation, we propose a new dam operation model based on a targeted operation plan, constrained by the regulation capacity of a dam and historical simulated streamflows, with flexibility to adjust to climate variation. The effects of dams on streamflows could then be studied with ORCHIDEE and isolated from the effect of climate factors and irrigation demands. Different from classical approaches separating the YRB into an upper, middle, and lower streams (Tang et al., 2008; Zhuo et al., 2019), we here further divide both the upper and middle streams into sub-catchments based on the locations of five

key gauging stations (Fig. 1). This approach splits regions with and without big dams (or large irrigation areas) in the upper and middle streams, which simplifies the assessment of the roles of irrigation and damming on streamflows.

In this study, ORCHIDEE with the novel crop-irrigation module (Wang, 2016; Yin et al., 2020) and the new dam operation model was applied in the YRB from 1982 to 2014 in order to: 1) demonstrate whether ORCHIDEE and the dam model, with generic parameterizations, are able to improve the simulation of streamflow fluctuations; and 2) attempt to separate the effect of

irrigation and dams on the fluctuations of monthly streamflows. We first describe ORCHIDEE model and our new dam model in Section 2.1. Then we present the algorithm used for estimating sub-catchment water balances in Section 2.2, followed by the input and evaluation datasets, the simulation protocol, and metrics for evaluation in Sections 2.3 to 2.5. Results are presented in Section 3 and limitations are discussed in Section 4.

## 2   Methodology

## 2.1   ORCHIDEE land surface model used in this study

### 2.1.1   Irrigation and crop modules

ORCHIDEE is a physical process-based land surface model that integrates hydrological cycle, surface energy balances, carbon cycle, and vegetation dynamics by two main modules. The SECHIBA (surface-vegetation-atmosphere transfer scheme) module simulates the dynamics of water cycle, energy fluxes, and photosynthesis at 0.5 hour time interval, which are used by the

STOMATE (Saclay Toulouse Orsay Model for the Analysis of Terrestrial Ecosystems) to estimate vegetation and soil carbon cycle at daily time step. The ORCHIDEE used in this study is a special version with newly developed crop and irrigation module (Wang et al., 2017; Wu et al., 2016; Yin et al., 2020). The crop module includes specific parameterizations for wheat, maize,

and rice, calibrated over China by observations (Wang, 2016; Wang et al., 2017). It is able to simulate crop carbon allocation, different phenological stages as well as related managements (e.g., planting date, rotation, multi-cropping, irrigation, etc).

Irrigation amount is simulated in the land surface model ORCHIDEE (Wang, 2016; Wang et al., 2017) as the minimum between crop water requirements and water supply. The crop water requirements are defined according to the choice of an irrigation technique, namely minimizing soil moisture stress for flooding technique, sustaining plant potential evapotranspiration for dripping technique, and maintaining the water level above the soil surface during specific months for paddy irrigation technique. Each crop is grown on a specific soil column (in each model grid-cell) where the water and energy budgets are in-

dependently resolved. The water resources in the hydrological routing scheme are from three water reservoirs: 1) a streamflow component; 2) a fast reservoir with surface runoff; and 3) a slow reservoir with deep drainage, used in this order for defining the priorities of water use for irrigation. As long-distance water transfer is not modeled, streams only supply water to the crops growing in the grid-cell they cross, according to the river routing scheme of the ORCHIDEE model (Ngo-Duc et al., 2007). Without dams, irrigation can be underestimated where dams stores water to supply the crop demand. Transfer from reservoirs,

lakes or local ponds to adjacent cells are not considered, which should further lead to an underestimation of the irrigation supply, dependent on the cell size. Details of the coupled crop-irrigation module of ORCHIDEE are described in Yin et al. (2020).

### 2.1.2  New dam operation model

To account for the impacts of dam regulation on streamflow ($Q$) seasonality, we developed a dynamic dam water storage

module based on only two generic rules: reducing flood peaks and guaranteeing baseflow. This model depends on simulated inflows and is thus independent from irrigation demands. It has been developed for the main reservoirs of the YRB (e.g., the LongYangXia, LiuJiaXia, and XiaoLangDi in Fig. 1). Different from Biemans et al. (2011); Hanasaki et al. (2006), we primarily consider the ability of reservoirs in regulating river flow seasonality. This means that the targeted baseflow and flood control of our dam model are not fixed proportions of the mean annual streamflow, but depends on the regulation capacity

of the reservoir ($C_{\mathrm{max}}$). Firstly, similar to Voisin et al. (2013), a multi-year averaged monthly streamflow ($Q_s$) is calculated based on ORCHIDEE simulations. To include the potential impacts of recent climate change on dam operation, here we only consider the latest past 10-year simulations, as:

$$Q_{\mathrm{s},i} = \frac{1}{N} \sum_{j}^{j \in N} Q_i^j. \tag{1}$$

Here $Q_{\mathrm{s},i}$ [m$^3$.s$^{-1}$] is a multi-year averaged monthly streamflow of month $i$; $j$ is a year index; $N$ is number of year accounted;

For a upcoming year $j$, we only use the historical simulations (maximum latest ten years) to calculate $Q_s$.

Secondly, we evaluate the targeted water storage change $\Delta W_t$ and monthly streamflow $Q_t$ considering the regulation capacity of each reservoir. As shown in Fig. S1, one year can be divided into two periods by comparing $Q_s$ with $\bar{Q}_s$. The longest continuous period of months with $Q_s > \bar{Q}_s$ is the recharging season for reservoirs, and the rest is the releasing season. The amount of water stored during the recharging season (blue region in Fig. S1) is determined by $C_{\mathrm{max}}$ and is used during the

releasing season (red regions in Fig. S1). The values of $\Delta W_t$ and $Q_t$ can be estimated by:

$$k = \min\left(\frac{C_{\max}}{\alpha \sum_{i}^{i \in \text{Recharge}} Q_{s,i}}, k_{\max}\right), \tag{2}$$

$$\Delta W_{t,i} = \alpha \left[ k \left( Q_{s,i} - \bar{Q}_s \right) + \bar{Q}_s \right], \tag{3}$$

$$Q_{t,i} = Q_{s,i} - \Delta W_{t,i}/\alpha. \tag{4}$$

Here $k$ [-], varying between 0 and $k_{\max}$ (=0.7), indicates the ability of reservoir in disturbing streamflow seasonality. It is a ratio

of the maximum regulation capacity of the reservoir $C_{\max}$ [$10^8$ m$^3$] over the streamflow amount throughout the recharging
season. $\alpha$ (0.0263) converts monthly streamflow to water volume. Assuming that the water storage of the reservoir reaches
$C_{\max}$ at the end of the recharging season, we can calculate targeted water storage $W_t$ by using $\Delta W_t$.

Finally, the variation of the actual water storage of the reservoir $\Delta W$ is a decision regarding actual monthly streamflow,
current water storage, $Q_t$, $\Delta W_t$, and $W_t$. During the releasing season, $\Delta W$ is calculated as:

$$\Delta W_i = \begin{cases} -W_i \dfrac{(-\Delta W_{t,i})}{W_{t,i}} & \text{if } W_i \le W_{t,i}; & \text{(5a)} \\[2mm] \Delta \tilde{W}_i - \left[ \left( W_i + \Delta \tilde{W}_i \right) - \left( W_{t,i} + \Delta W_{t,i} \right) \right] & \text{if } W_i > W_{t,i} \text{ and } \Delta \tilde{W}_i > \Delta W_{t,i}; & \text{(5b)} \\[2mm] \Delta W_{t,i} - \left( W_i - W_{t,i} \right) & \text{if } W_i > W_{t,i} \text{ and } \Delta \tilde{W}_i \le \Delta W_{t,i}. & \text{(5c)} \end{cases}$$

Here $\Delta \tilde{W}_i = \alpha Q_i - (\alpha Q_{t,i} - \Delta W_{t,i})$. It is the expected release amount to make river streamflows equal to the targeted stream-
flows after reservoir regulation. If current water storage is less than the targeted value (the case of Eq. 5a), the $\Delta W_i$ is calculated
by the $W_i$ with a proportion of $\Delta W_{t,i}$ over $W_{t,i}$. If the current storage is more than the targeted value (the cases of Eq. 5b and
5c), the reservoir can release more water based on a balance between the targeted water storage change $\Delta W_{t,i}$ and the tar-

geted water storage at the next time step $W_{t,i}$ (represented by $\Delta \tilde{W}_i$). Note that all water storage change variables are negative
throughout the releasing season.

During the recharging season, we can calculate the $\Delta W_i$ as:

$$\Delta W_i = \begin{cases} \max\left(\min\left(W_{t,i} + \Delta W_{t,i} - W_i, \alpha Q_i\right), 0\right) & \text{if } W_i > W_{t,i}; & \text{(6a)} \\[2mm] \min\left(\Delta W_{t,i} + \left(W_{t,i} - W_i\right), \alpha Q_i\right) & \text{if } W_i \le W_{t,i}. & \text{(6b)} \end{cases}$$

If current water storage is larger than the targeted value (Eq. 6a), we will try to recharge a volume of water to make $W_{i+1} =$

$W_{t,i+1}$. If current water storage is less than the targeted value (Eq. 6b), we decide to recharge additional water volume besides
the $\Delta W_{t,i}$.

$\Delta W$ is then applied as a correction of simulated streamflows to generate actual monthly streamflows using the following
equation:

$$\hat{Q}_{\text{sim},i} = Q_{\text{sim},i} - \frac{1}{\alpha} \Delta W_i. \tag{7}$$

Here $\hat{Q}_{\text{sim}}$ [m³.s⁻¹] is the simulated regulated streamflows; $Q_{\text{sim}}$ [m³.s⁻¹] is the simulated monthly streamflows. Note that this model is a simplified representation of dam management, because it ignores the direct coupling between water demand and irrigation water supply from the cascade of upstream reservoirs. This approach implies that, with a regulated flow, demands will be able to be satisfied and floods to be avoided without being more explicit. A complete coupling of demand, flood, and reservoir management is difficult to implement in the land surface model in absence of data about the purpose and management
strategy of each dam, given different possibly conflicting demand of water for industry and drinking versus cropland irrigation.

Before performing the simulation, we estimate the maximum regulation capacity of each studied reservoir in each river sub-catchment shown in Fig. 1. Table 1 lists collected information of the main reservoirs on the YR. Only large reservoirs like LongYangXia (LYX), LiuJiaXia (LJX), and XiaoLangDi (XLD) are considered in our model because of their huge $C_{\text{max}}$.

## 2.2 Sub-catchment diagnosis

Figure 1 shows the YRB and main gauging stations used in this study. To effectively use $Q_{\text{obs}}$ for investigating impacts of irrigation and dam regulations on the streamflows of different river sub-catchments, we divided the YRB into five sub-catchments ($R_i$, $i \in [1,5]$, Fig. 1) with an outlet at each gauging station. Thus we can evaluate the water balance in $R_i$ by:

$$\frac{\Delta \text{TWS}_i}{\Delta t} = P_i - \text{ET}_i + \frac{Q_{\text{in},i} - Q_{\text{out},i}}{A_i}. \tag{8}$$

Where $\Delta t$ is time interval; $\Delta \text{TWS}_i$ [mm] is change of total water storage in specific $R_i$; $P_i$ [mm.$\Delta t^{-1}$] is precipitation in $R_i$; $\text{ET}_i$ [mm.$\Delta t^{-1}$] is evapotranspiration in $R_i$; $A_i$ [m²] is area of $R_i$. $Q_{\text{in},i}$ and $Q_{\text{out},i}$ [m³.$\Delta t^{-1}$] are inflow and outflow respectively. In addition, $q_i = Q_{\text{out},i} - Q_{\text{in},i}$ indicates the contribution of $R_i$ to the river streamflows, that is the sub-catchment streamflows. This term can be negative if local water supply (e.g., precipitation and groundwater) cannot meet water demand. A conceptual figure of the water balance of a sub-catchment is shown at the top left of Fig. 1.

## 2.3 Evaluation datasets


Observed monthly streamflows ($Q_{\text{obs}}$) from the gauging stations shown in Fig. 1 are used to evaluate the simulations. Several precipitation ($P$) and evapotranspiration (ET) datasets were selected to evaluate the simulated water budgets in each sub-catchment of the YRB. The 0.5° 3-hourly precipitation data from GSWP3 (Global Soil Wetness Project Phase 3) used as model input is based on GPCC v6 (Global Precipitation Climatology Centre (Becker et al., 2013)) after bias correction with ob-
servations. The MSWEP (Multi-Source Weighted-Ensemble Precipitation) is a 0.25° 3-hourly $P$ product integrating numerous in-situ measurements, satellite observations, and meteorological reanalysis (Beck et al., 2017). Three ET datasets are chosen for their potential ability to capture the effect of irrigation disturbance on ET (Yin et al., 2020) (noted as $\text{ET}_{\text{obs}}$). GLEAM v3.2a (Global Land Evaporation Amsterdam Model, (Martens et al., 2017)) provides 0.25° daily ET estimations based on a two-soil layer model in which the top soil moisture is constrained by the ESA CCI (European Space Agency Climate Change Initia-
tive) Soil Moisture observations. The FLUXCOM model (Jung et al., 2009) upscales ET data from a global network of eddy covariance towers measurements into a global 0.5° monthly ET product. Since these towers do not cover irrigated systems, ET

from irrigation simulated by the LPJml (Lund-Postam-Jena managed Land) is added to ET from non-irrigated systems. The PKU ET product estimates 0.5° monthly ET by water balances at basin scale integrating FLUXNET observations to diagnose sub-basin patterns by a Multiple Tree Ensemble approach (Zeng et al., 2014).

## 2.4 Simulation protocol

The 0.5° half-hourly GSWP3 atmospheric forcing (Kim, 2017) was used to drive ORCHIDEE simulations. Yin et al. (2018) used four atmospheric forcing to drive ORCHIDEE to simulate soil moisture dynamics over China and they found that the GSWP3 provided the best performances, hence we chose this forcing for this sutdy. A 0.5° map with 15 different Plant Functional Types (PFTs) containing crop sowing area information for the three PFTs corresponding to the modeled crop (wheat, maize, and rice) is used, based on 1:1 million vegetation map and provincial scale census data of China. Crop planting dates for wheat, maize, and rice are derived from spatial interpolation of phenological observations from Chinese Meteorological Administration (Wang et al., 2017). Soil texture map is from Zobler (1986). Two simulation experiments were performed to assess the impacts of irrigation on streamflows: 1) NI: no irrigation; 2) IR: irrigated by available water resources. In IR, only surface irrigation is considered, that is water applied on the cropland soil without interception by canopies. The soil water stress, a function of soil moisture and crop root density up to 2 m depth (Yin et al., 2020), is checked every half an hour. When it is less than a target threshold, irrigation is triggered with amount equal to the difference between saturated and current soil moisture. To precisely estimate irrigation water consumption (direct water loss from the surface water pool excluding return flow), deep drainages of the three crop soil columns is turned off in the IR simulation. Simulations start from a 20-year spin-up in 1982 to initialize the thermal and hydrological variables, then continued from 1982 to 2014. A validation against naturalized streamflows is shown in Table S1.

The dam operation simulation starts from 1982 as an offline model applied to the simulated streamflows from the IR simulation ($Q_{IR}$) as input. The initial values of $W$ were set to half of the $C_{max}$. Considering potential joint regulation of reservoirs, we firstly estimate the total $\Delta W$ of all considered reservoirs by using $Q_{IR}$ at HuaYuanKou (outlet of R$_4$, Fig. 1). Then we estimate the $\Delta W$ of LYX and LJX reservoir by using $Q_{IR}$ at LanZhou. The difference between these two $\Delta W$ is assumed to be the $\Delta W$ of the XLD reservoir in-between. Offline simulated $\Delta W$ values are used to estimate regulated monthly streamflows ($\hat{Q}_{IR}$) as Eq. 7. As huge irrigation water withdrawals occur in R$_3$ and R$_5$ (YRCC, 1998–2014) the water recharge of reservoirs may result in negative $\hat{Q}_{IR}$ at TouDaoGuai and LiJin. To avoid this numerical artifact due to the offline nature of our dam model, we corrected all negative $\hat{Q}_{IR}$ to zero by assuming that the streamflows cannot further drop when all stream water is consumed upstream. The impact of this corrections are accounted for at gauging stations downstream to ensure mass conversation.

## 2.5 Evaluation metrics

Three metrics are used to evaluate the performances of simulated monthly $Q$. The mean-square error (MSE) evaluates the magnitude of errors between simulation and observations. It can be decomposed into three components (Kobayashi and Salam,

2000):

$$\text{MSE} = \frac{1}{n}\sum_{i=1}^{n}(S_i - O_i)^2 = \text{SB} + \text{SDSD} + \text{LCS}. \tag{9}$$

Where $S_i$ and $O_i$ are simulated and observed values, respectively; $n$ is the number of samples. SB (squared bias) is the bias between simulated and observed values. In this study, SB represents the difference between simulated and observed multi-year mean annual $Q$. SDSD (the squared difference between standard deviation) relates to the mismatch of variation amplitudes between simulated and measured values. It can reflect whether our simulation can capture the seasonality of $Q_{\text{obs}}$. LCS (the lack of correlation weighted by the standard deviation) indicates the mismatch of fluctuation patterns between simulated and

observed values, which is equivalent to inter-annual variation of $Q$ in this study. The formulas of these three components and detailed explanation can be found in Kobayashi and Salam (2000).

     The index of agreement ($d \in [0,1]$) is defined as the ratio of MSE and potential error. It is calculated as:

$$d = 1 - \frac{\sum_{i=1}^{n}(O_i - S_i)^2}{\sum_{i=1}^{n}\left(|S_i - \bar{O}| + |O_i - \bar{O}|\right)^2}. \tag{10}$$

$d = 1$ indicates perfect fit, while $d = 0$ denotes poor agreement.

The modified Kling-Gupta Efficiency (mKGE $\in (-\infty, 1]$) is defined as the Euclidean distance of three independent criteria: correlation coefficient $r$, bias ratio $\beta$, and variability ratio $\gamma$ (Gupta et al., 2009; Kling et al., 2012). It is an improved indicator from the Nash-Sutcliffe Efficiency avoiding heterogeneous sensitivities to peak and low flows, which is crucial for this study that is not only interested in simulating peak flows but also concentrates on base flows regulated by dams for human usage. mKGE is calculated as,

$$\text{mKGE} = 1 - \sqrt{(1-r)^2 + (1-\beta)^2 + (1-\gamma)^2}, \tag{11}$$

$$\beta = \frac{\mu_S}{\mu_O}; \gamma = \frac{\text{CV}_S}{\text{CV}_O}, \tag{12}$$

where $r$ is the correlation coefficient between observed and simulated streamflows; $\mu$ [m$^3$.s$^{-1}$] and CV [-] are the mean and the coefficient of variation of $Q$, respectively. These indicators are used for three comparisons: 1) $Q_{\text{NI}}$ and $Q_{\text{obs}}$; 2) $Q_{\text{IR}}$ and

$Q_{\text{obs}}$; 3) $\hat{Q}_{\text{IR}}$ and $Q_{\text{obs}}$.

## 3    Results

### 3.1    Water budgets at sub-catchment scale

Figure 2 displays water budgets and trends in R$_i$ based on simulation and observations. Going from upstream to downstream, precipitation in $P_{\text{GSWP3}}$, which is consistent with $P_{\text{MSWEP}}$, decreases from 543.6 mm.yr$^{-1}$ (R$_1$) to 254.2 mm.yr$^{-1}$ (R$_3$), and

then rises again to 652.1 mm.yr$^{-1}$ (R$_5$). The magnitudes of simulated ET (both ET$_{\text{NI}}$ and ET$_{\text{IR}}$) have no significant differences

with $ET_{obs}$ aggregated over sub-catchments $R_1$ to $R_5$. Grid cell-based validation shows high agreement between simulated and observed ET across all sub-catchments. The lowest mean of correlation coefficients is 0.79 and the highest mean of relative RMSE is 4.9% (Table S2). Except for $R_1$ where cropland is rare, $ET_{IR}$ accounts for an amount representing more than 80% of $P_{GSWP3}$ in the YRB, with a maximum value of 96.5% in $R_3$. The difference between $ET_{IR}$ and $ET_{NI}$ is due to the irrigation process, which accounts for 9.1% and 8.2% of $ET_{NI}$ in $R_3$ and $R_5$ respectively as caused by the irrigation demand. The impact of irrigation can be detected from sub-catchment streamflows ($q_i = (Q_{out,i} - Q_{in,i})/A_i$) as well. For instance, both $q_{obs}$ and $q_{IR}$ are negative in $R_3$ and $R_5$, suggesting that local surface water resources cannot meet water demand for irrigation. As irrigation water transfers between grid cells are not represented in our simulations, the non-availability of water locally results in an underestimation of the irrigation withdrawals, likely explaining why $q_{IR} > q_{obs}$ in $R_3$ to $R_5$.

The trends of $P$ and ET are positive but not significant in most $R_i$ during the period 1982–2014 (bottom panel of Fig. 2). However, significant trends can be found in simulated and observed $q$ in some $R_i$. The decrease of $q_{obs}$ in $R_1$ is not captured by the model, neither in $q_{NI}$ nor $q_{IR}$. This underestimated decrease of river streamflows might be linked to decreased glacier melt or increased non-irrigation human water withdrawals, which are ignored in our simulations. In $R_2$ and $R_3$, the $q_{obs}$ trends are determined by the joint effects of climate change (e.g., the $P$ increase) and human water withdrawals. The trends of $q_{IR}$ show the same direction as that of $q_{obs}$. In $R_5$ however $q_{obs}$ increased by $1.67 \, \text{mm.yr}^{-1}$, which was not captured by our simulation of $q_{IR}$. Besides the increase of $P$, another possible driver of increasing $q_{obs}$ in $R_5$ is a decrease of water withdrawal due to the improvement of irrigation efficiency (Yin et al., 2020), which is not accounted in our simulations. Moreover, the water use management may play an important role in the observed positive trends of $q_{obs}$ as well, with the aim to increase the streamflows at the downstream of the YR to avoid streamflow cutoff ($Q_{obs} < 1 \, \text{m}^3.\text{s}^{-1}$) that occurred in 1990's (Wang et al., 2006).

Irrigation not only influences annual streamflows in the YR, but also affects its intra-annual variation. In general, the discharge yield $Y_Q$, defined by the sum of surface runoff and drainage, of all grid cells in NI should be higher than in IR because our irrigation model can remove water from the stream reservoirs which is a fraction of drainage and runoff. However, our simulations show that $Y_{Q,NI}$ can be less than $Y_{Q,IR}$ (Fig. S2) at the beginning of the monsoon season. This is because irrigation keeps soil moisture higher than without irrigation in July in $R_4$ and $R_5$ (Fig. S2d and S2e), which in turn promotes $Y_Q$ because the soil water holding capacity is lower and a larger fraction of $P$ can go to runoff. This mechanism highlighted that irrigation could enhance the heterogeneity of water temporal distribution and may reinforce floods after a dry season.

### 3.2 Comparison between observed and simulated $Q$

Figure 3 shows time series of annual streamflows and of the seasonality of monthly streamflows. Our simulations underestimate $Q_{obs}$ at TangNaiHai in $R_1$ likely because we miss glacier melt. After LanZhou, the values of $Q_{IR}$ coincide very well with those of $Q_{obs}$, indicating that irrigation strongly reduces the annual streamflows of the YR. However, the seasonality of monthly $Q_{IR}$ is different from $Q_{obs}$ (Fig. 3f-3j). Despite the good match of annual values, the model without dams (shown in Fig. ?? produces an underestimation of $Q$ in the dry season and an overestimation of $Q$ in flood season. Such a mismatch of $Q$ seasonality is likely caused primarily by dam regulation ignored in the model. The locations of several big reservoirs are shown in the bottom panel of Fig. 1 and their characteristics are listed in Table 1.

Regarding dams, before the operation of the LongYangXia dam in 1986 which brought a regulation capacity of $193.5 \times 10^8 \, \mathrm{m}^3$ (green bar in Fig. 4b), the peaks of monthly $Q_{\mathrm{NI}}$ at LanZhou were slightly lower than the peaks of $Q_{\mathrm{obs}}$ in $R_2$ (Fig. 4b), as well as at TangNaiHai (Fig. 4a). But after the construction of the LongYangXia dam, modeled peak $Q_{\mathrm{NI}}$ became systematically higher than the peak of $Q_{\mathrm{obs}}$ each year, suggesting that the construction of this dam caused the observed peak reduction (Fig. 4b). Moreover, the seasonality of $Q_{\mathrm{obs}}$ changed dramatically in the period (1982-2014), but no similar trend was found in monthly $P$ (Fig. S3), suggesting that dam operation was the primary driver of the observed shift in seasonal streamflow variations of the YRB from 1982 to 2014. Dams can affect inter-annual variations of $Q$ as well, although less than the seasonal variation. For instance, TongGuan and XiaoLangDi are two consecutive gauging stations upstream and downstream the reservoir of XiaoLangDi in $R_4$ (Fig. 1). The annual $Q_{\mathrm{obs}}$ at the two stations shows different features after the construction of the XiaoLangDi reservoir in 1999.

Figure 5 shows monthly time series of $Q_{\mathrm{obs}}$, $Q_{\mathrm{IR}}$, and $\hat{Q}_{\mathrm{IR}}$ (see Sect. 2.1.2) at each gauging station. Discharge fluctuations are successfully improved in $\hat{Q}_{\mathrm{IR}}$. Especially the baseflow of $\hat{Q}_{\mathrm{IR}}$ coincides well with that of $Q_{\mathrm{obs}}$ during winter and spring. The only exception occurs at LiJin, where $\hat{Q}_{\mathrm{IR}}$ overestimates the streamflows from January to May. In fact, the water release from XLD during this period would be withdrawn for irrigation and industry in $R_5$. However, our offline dam model is not able to simulate the interactions, leading to the overestimation.

The dam model is successful to reproduce flood control as well. At LanZhou, although $\hat{Q}_{\mathrm{IR}}$ underestimates the peak flow due to the bias of the simulated mean annual streamflows (Fig. 3b), its seasonality is much smoother than that of $Q_{\mathrm{IR}}$. The underestimation of $\hat{Q}_{\mathrm{IR}}$ can reflect special water management during extreme years. From 2000 to 2002, the YRB experienced severe droughts with 10~15% precipitation less than usual, leading to a decrease of surface water resource as much as 45% (Water Resources Bulletin of China, http://www.mwr.gov.cn/sj/tjgb/szygb/). To guarantee base flow, a set of policies were applied (e.g., reducing water withdrawn, increasing water price, releasing more water from reservoirs). Those policies are not accounted for the model, which will produce a higher irrigation demand during dry years and promote the underestimation of the $Q_{\mathrm{obs}}$. From TouDaoGuai to LiJin, the floods from August to October are dramatically reduced by our dam model. Nevertheless, the peaks are still overestimated in $\hat{Q}_{\mathrm{IR}}$, which might be due to numerous non-modeled medium/small reservoirs that were ignored by our model, no less than 203 medium reservoirs were documented at the end of 2014 (YRCC, 1998–2014). In our simulation, a $326.5 \times 10^8 \, \mathrm{m}^3$ regulation capacity is considered, which only accounts for 45% of the total storage capacity of $720 \times 10^8 \, \mathrm{m}^3$ (Ran and Lu, 2012). Moreover, in the five irrigation districts (http://www.yrcc.gov.cn/hhyl/yhgq/, (Tang et al., 2008)), special irrigation systems in the YRB, could contribute to flood reduction. For instance, the Hetao Plateau is the traditional irrigation district and is equipped with a hydraulic system that can divert river water into a complex irrigation network, [106.5–109°E]×[40.5–41.5°N] in Fig. 1, by adjusting water level differences during the flood season. This no-dam diversion system of the Hetao Plateau can take $50 \times 10^8 \, \mathrm{m}^3$ as an extra regulation capacity per year, equivalent to 14% of the annual streamflow in $R_3$.

Simulated $\Delta W$ in $R_2$ is compared to observations (Jin et al., 2017) in the left panel of Fig. 6, and the good agreement suggests that our dam model is able to capture the seasonal variation of $\Delta W$ ($r = 0.9$, $p < 0.001$) and rectify the simulated streamflows. In the case of XiaoLangDi in the right panel of Fig. 6, where the correlation is smaller ($r = 0.34$, $p = 0.28$), the

335 mismatch could be explained by sediment regulation procedures of that dam, given that it releases a huge amount of water in June for reservoir cleaning and sediment flushing downstream (Baoligao et al., 2016; Kong et al., 2017; Zhuo et al., 2019), a process not represented in our simple dam model. Moreover, because we ignored the buffering effect of numerous medium reservoirs, the simulated water recharge during the flood season could be overestimated.

Figure 7 presents the model performances with different metrics in different $R_i$. The results show that MSE increases 340 considerably from $R_1$ to $R_5$, implying accumulated impacts of error sources in increasing the error of modeled $Q$ when going downstream in the entire catchment. Most likely those error sources are omission errors of anthropogenic factors such as drinking and industrial water removals, but also of natural factors such as riparian wetlands and floodplains (e.g., the SanSheng-Gong water conservancy hub), and non-represented small streams in the routing of ORCHIDEE (e.g., the irrigation system at the Hetao Plateau). From the decomposition of MSE, we found that adding irrigation to the model removes most of the bias 345 in the average magnitude $Q$ by reducing the SB bias error term of the MSE. The only exception occurs at LanZhou, where SB increases in IR consequently leading to higher MSE. This misfit is due to the underestimation of $Q$ upstream (Fig. 3a). Thus, modeled $Q_{IR}$ is lower at LanZhou which enlarges the SB. On the other hand, adding the dam operation contributes to improve the phase variations of $Q$ which are dominated by the phase of the seasonal cycle, by reducing the SDSD error term. Nevertheless, the LCS error term indicating the magnitude of the variability, mainly at inter-annual time scales, has no 350 significant improvement with the representation of irrigation and dam regulations. It is because some of reservoirs are able to regulate $Q$ inter-annually (Table 1), which can be observed from Fig. 4c. However, related operation rules are unclear and are not implemented in our dam model. Improvements were found in $d$ as well, which demonstrates that the way human effects on $Q$ of the YR were modeled brings more realistic results, despite ignoring the direct effect of irrigation demand on reservoir release, and ignored industrial and domestic water demands. The mKGE reveals significant increase after considering dam 355 operations (Fig. 7d). Particularly at LanZhou and HuaYuanKou, the mKGE of $\hat{Q}_{IR} \sim Q_{obs}$ increases 0.86 and 1.11 than that of $Q_{IR} \sim Q_{obs}$, respectively. Note that the mKGEs of $Q_{IR} \sim Q_{obs}$ are smaller than that of $Q_{NI} \sim Q_{obs}$ from $R_2$ to $R_4$, because irrigation decreases the mean annual streamflow of $Q_{IR}$, which further increases the $CV_S$ leading to worse $\gamma$ in mKGE (Eq. 11).

## 4 Discussion

This study shows that ORCHIDEE land surface model with crops, irrigation, and our simple dam operation model can repro-360 duce correctly streamflow mean levels, inter-annual variations, and seasonal cycle in different sub-catchments of the YRB. We preliminarily quantified the impacts of irrigation and dams on the fluctuations of streamflows. Simulated water balance components were compared to observations in different sub-catchments with a good agreement (e.g., $4.5 \pm 6.9\%$ for ET). We found that irrigation mainly affects the magnitude of annual streamflows by consuming $242.8 \pm 27.8 \times 10^8 \, \mathrm{m^3.yr^{-1}}$ of water, consistent with census data giving a consumption of $231.4 \pm 31.6 \times 10^8 \, \mathrm{m^3.yr^{-1}}$ (YRCC, 1998–2014). As the water of the 365 YRB is reaching the limit of usage (Feng et al., 2016), we did not find any significant effect of irrigation on streamflow trends. Instead of increasing river water withdrawals, the growing water demand appeared to have been balanced by improving water use efficiency during the study period (Yin et al., 2020; Zhou et al., 2020). Our simulation reveals that the impact of irrigation

on streamflows may even be positive under special situations, which was also shown in Kustu et al. (2011). However, our mechanisms are different from the irrigation-ET-precipitation atmospheric feedback mechanisms found by Kustu et al. (2011), we demonstrated that irrigation may significantly increase soil moisture and promote runoff yield during the following wet season. It implies that irrigation in such landscapes may reinforce the magnitude of floods during the rainy season by a higher legacy soil moisture.

We found that dams strongly regulate the temporal variation of streamflows (Chen et al., 2016; Li et al., 2016; Yaghmaei et al., 2018). By including simple regulation rules depending on inflows, our dam model reduced by 48–77% the simulation error (MSE in Fig. 7), especially thevariability component (SDSD) of the total error which is dominated by seasonal misfit reduction from dams. Moreover, we confirmed that the change of $Q_{obs}$ seasonality during the study period is not due to climate change (Fig. S3), but is determined by dam operations (Wang et al., 2006). Big dams, like the LongYangXia, LiuJiaXia, and XiaoLangDi, are able to regulate streamflows inter-annually (Wang et al., 2018) and smooth the inter-annual distribution of water resources in YRB (Piao et al., 2010; Wang et al., 2006; YRCC, 1998–2014). However, their detailed operation rules are unclear and were not implemented explicitly in our dam model. The error corresponding to inter-annual variation (LCS in MSE in Fig. 7) was thus not reduced by including dams. In the dam model, some functions of reservoirs, such as providing irrigation supply, industrial and domestic water, electricity generation, and flood control (Basheer and Elagib, 2018) are not explicitly represented. Particularly the XiaoLangDi dam carries a distinctive water-sediment mission, which scours sediments downstream by creating artificial floods in June (Kong et al., 2017; Zhuo et al., 2019). These functions are associated with many socioeconomic factors and drivers leading to competing demands for water (e.g., policies, electricity price, water price, land use change, irrigation techniques, water management techniques, and dams inter-connection), which are difficult to be well modeled due to lack of data. However, with the upcoming Surface Water and Ocean Topography (SWOT) mission, it will be possible to monitor the water level and surface extent of more reservoirs, which will be helpful to improve and validate the dam operation simulations (Ottlé et al., 2020).

Our simulations ignored potential impacts of reservoirs on local climate, e.g. through their evaporation (Degu et al., 2011). The water area of several artificial reservoirs (LongYangXia, LiuJiaXia, BoHaiWan, SanShengGong, and XiaoLangDi) is approximately $1056\,km^2$, which is larger than the 10th largest natural lake in China. These water bodies can also significant influence local energy budgets and evaporative water loss may be considerable especially in arid and semi-arid area (Friedrich et al., 2018; Shiklomanov, 1999). Besides, the five large irrigation districts (http://www.yrcc.gov.cn/hhyl/yhgq/) could dramatically alter the local climate through atmospheric feedbacks. For instance, the Hetao Plateau can take about $50\times10^8\,m^3$ from streamflows every year during the flood season. Its irrigation area is $5740\,km^2$ with an evapotranspiration rate ranging between $1200{\sim}1600\,mm.yr^{-1}$. However, as these irrigation districts divert river water without big dams, they are not taken into account in most YR studies. Another non-negligible factor in the case of YR is sedimentation, which reduces the regulation capacities of reservoirs and weakens streamflow regulation by human. For instance, the total capacity of QingTongXia declined from 6.06 to $0.4\times10^8\,m^3$ since 1978 due to sedimentation. Therefore, how land use change and evolution of natural ecosystems affect sediment load and deposition is another key factor to project dams disturbances on streamflows in the YRB.

Simulating anthropogenic impact on river streamflows is challenging. In the case of the YR, well calibrated models can provide accurate naturalized streamflow simulations with Nash-Sutcliffe Efficiency (NSE) as high as 0.9 (Yuan et al., 2016). However, when considering the impacts of irrigation and dams, the NSE values of simulations are generally worse. For instance, simulations with anthropogenic effects by Hanasaki et al. (2018) had lower NSE than the simulation with only natural processes. Similarly, Wada et al. (2014) showed NSE decrease after considering anthropogenic factors in the YRB, which was interpreted as complexity of the YRB under the impacts of human activities and climate variation. However, the NSE of naturalized streamflows cannot really be compared to the one of regulated streamflows. Even if a model can perfectly simulate the dam operations, the NSE of naturalized streamflows will be larger than that of regulated streamflows, given that dam operations automatically reduce the variation of river streamflows (a simple proof is available in Sect. A of the online supplement). In fact, our model performances are very similar to GHM simulations (Fig. S2 from Liu et al. (2019)). By gradually considering anthropogenic factors (irrigation and dam operations) the performances of our simulations increase dramatically according to all the three metrics.

Intensive calibrations using a suite of observations can allow catchment-scale studies to provide high-accurate simulated streamflows for short-term flood forecast. However, for long-term projections, a model should include all key processes of the system studied. If key processes are missing in the model, a calibration will cover up the shortcomings, which lead to lack of predictive capacities for long time scales as shown by Duethmann et al. (2020). Therefore, by developing crop physiology and phenology, irrigation, and (offline) dam operation model, we have tried to demonstrate that streamflow fluctuations of the YR can be reasonable reproduced by a generic land surface model. Although mismatches exist in the simulated streamflows, they are more likely caused by missing processes (joint impact of multiple medium reservoirs, special mission of dams, irrigation system characteristics) than by poor calibration of existing processes, because other simulated hydrological variables coincide well with observations in the YRB, such as soil moisture dynamics (Yin et al., 2018), naturalized river streamflows (Table S1 in Xi et al. (2018)), leaf area index (Section S2 in Xi et al. (2018)), amount and trend of irrigation withdrawals (Yin et al., 2020), trends of total water storage (Section 3.4 in Yin et al. (2020)), and ET (Table S2). On the contrary, these mismatches draw our attention to some key mechanisms overlooked in most models. For instance, our model underestimates the annual streamflow at LanZhou in the period 2000–2002 (Fig. 3b), during which $\hat{Q}_{IR}$ was almost negatively correlated to the $Q_{obs}$ (Fig. 5a). In summary, our results show that the errors of simulated streamflows decreased dramatically after considering crops, irrigation, and dam operations, suggesting that these are first order mechanisms controlling streamflow fluctuations. Future work can be focused on completing the model by linking dam operation to the variable crop water demand.

## 5 Conclusions

A land surface model ORCHIDEE and a newly developed dam model are utilized to simulate the streamflow fluctuations and dam operations in the Yellow River Basin. The impacts of irrigation and dam regulation on streamflow fluctuations of the Yellow River were preliminarily qualified and quantified in this study by using a process-based land surface model and a dam operation model. Irrigation mainly contributes to the reduction of annual streamflow by as much as $242.8 \pm 27.8^8 \, \mathrm{m}^3.\mathrm{yr}^{-1}$.

The shifts of intra-annual variation of the Yellow River streamflows appear not to be caused by climate change, at least not by significant changes of precipitation patterns and land use during the study period, but by the construction of dams and their operation. After considering the impacts of dams, we found that dam regulation can explain about 48–77% of the fluctuations of streamflows. The effect of dams may be still underestimated because we only considered simple regulation rules based on inflows, but ignored its interactions with irrigation demand downstream. Moreover, our analysis showed that several reservoirs on the Yellow River are able to influence streamflows inter-annually. However, such effects are not quantified due to lack of knowledge of the regulation rules across our study period.

*Code and data availability.* The code of ORCHIDEE can be assessed via https://forge.ipsl.jussieu.fr/orchidee/wiki. The data used in this study, and the code of the dam operation model, analysis, and plotting can be accessed via https://doi.org/10.5281/zenodo.3979053 (Yin, 2020). The GLEAM ET data can be downloaded from http://gleam.eu (Martens et al., 2017). The MSWEP precipitation data and the PKU ET are available from http://gloh2o.org (Beck et al., 2017) and Zhenzhong Zeng (Zeng et al., 2014), respectively, which can be obtained upon reasonable requests.

*Author contributions.* ZY, CO, and PC designed this study; ZY and XW contributed to the model developments; ZY, FZ, XW, XZ, YB, and YX prepared observed datasets; ZY performed model simulations and primary analysis, and drafted the manuscript; all authors contributed to results interpretation, additional analysis, and manuscript revisions.

*Competing interests.* The authors declare that they have no conflict of interest.

*Acknowledgements.* We would like to thank the editor and the two anonymous referees for their insightful comments and efforts. We gratefully acknowledge the GLEAM team, the FLUXCOM team, Hyungjun Kim, Hylke Beck, and Zenzhong Zeng for their selfless sharing of their datasets.

*Financial support.* This study was supported by the National Natural Science Foundation of China (grant number 41561134016), the CHINA-TREND-STREAM French national project (ANR Grant No. ANR-15-CE01-00L1-0L). This work was also supported by the French Space Agency (Centre National d'Etudes Spatiales) through the TOSCA-SPAWET program in preparation to the international SWOT space mission. Zhou F. was supported by the National Key Research and Development Program of China (2016YFD0800501).

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

**Table 1.** Information of artificial reservoirs on the YR with considerable total capacity. Data is mainly from the YR Conservancy Commission of the Ministry of Water Resources (http://www.yrcc.gov.cn). The "regulation purposes" follows the style of Hanasaki et al. (2006). "H" indicates hydropower; "C" indicates flood control; "I" indicates irrigation; "W" indicates water supply; and "S" indicates scouring sediment.

| Name | Total capacity ($10^8$ m$^3$) | Regulation capacity ($10^8$ m$^3$) | Regulation since | Regulation type | Regulation purposes |
|---|---|---|---|---|---|
| LongYangXia | 247 | 193.53 | Oct 1986 | Inter-annual | HCIW |
| LiJiaXia | 16.5 | – | Dec 1996 | Daily, weekly | HI |
| GongBoXia | 6.2 | 0.75 | Aug 2004 | Daily | HCIW |
| LiuJiaXia | 57 | 41.5 | Oct 1968 | Inter-annual | HCIW |
| YanGuoXia | 2.2 | – | Mar 1961 | Daily | HI |
| BaPanXia | 0.49 | 0.09 | – | Daily | HIW |
| QingTongXia | $6.06 \rightarrow 0.4^*$ | – | 1968 | Daily | HI |
| XiaoLangDi | 126.5 | 91.5 | 1999 | Inter-annual | CSWIH |

$^*$ The total capacity shrink is due to sedimentation.

**Table 2.** Definitions of sub-catchments and values of $C_{dam}$ used in the dam regulation simulation.

| Sub-catchment | Stations | $C_{dam}$ ($10^8$ m$^3$) | Regulation since |
|---|---|---|---|
| R$_1$ | –<br>TangNaiHai | – | – |
| R$_2$ | TangNaiHai<br>LanZhou | 41.5 before 1987;<br>235 after 1987 | 1982 |
| R$_3$ | LanZhou<br>TouDaoGuai | – | 1982 |
| R$_4$ | TouDaoGuai<br>HuaYuanKou | 91.5 | 1999 |
| R$_5$ | HuaYuanKou<br>LiJin | – | – |

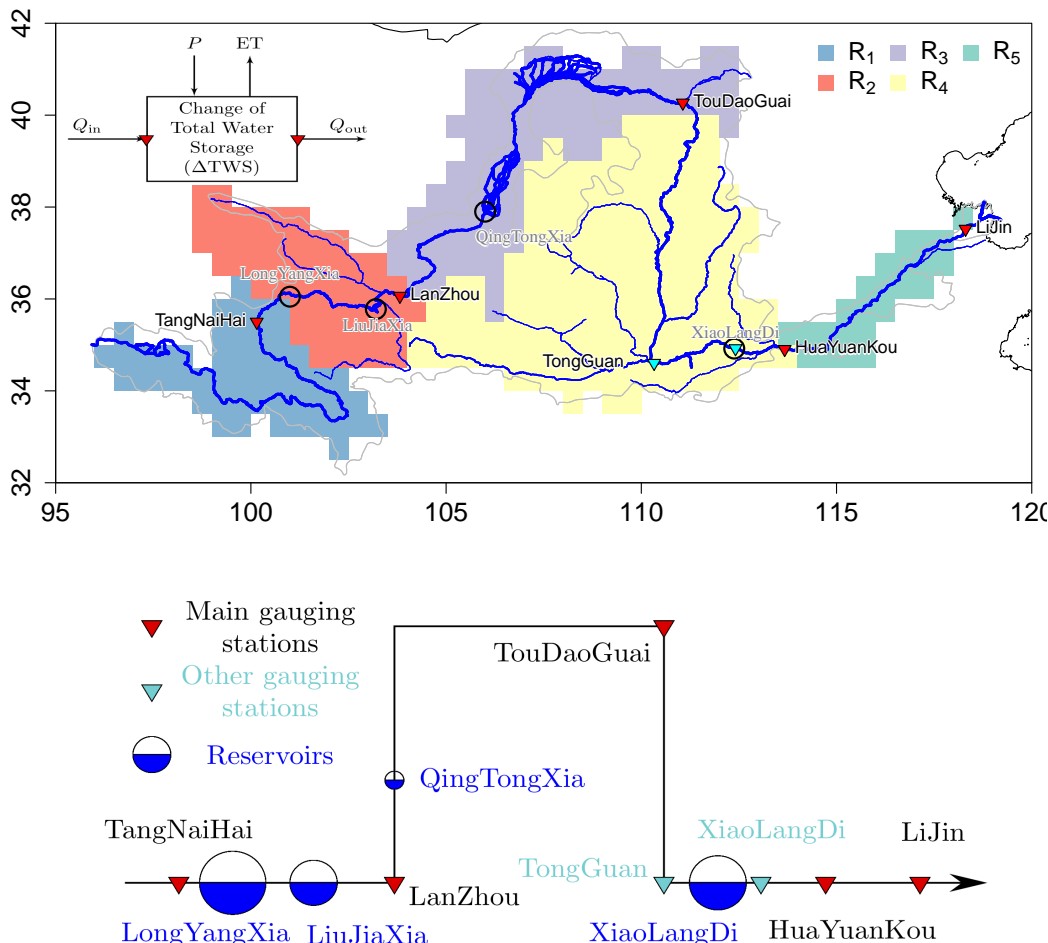

**Figure 1.** Top panel: map of YRB. Gray and blue lines indicate the catchment and network of YR based on GIS data, respectively. Dark circles are main artificial reservoirs on the YR. Triangles are gauging stations. Red triangles are main stations used for classifying sub-catchment and simulation comparison, and teal triangles are stations used to assess the impacts of XiaoLangDi Reservoir on river streamflows. Colored patterns are sub-catchments between two neighbouring gauging stations based on ORCHIDEE routing map. The water balances of specific sub-catchment are shown at the top left. Bottom panel: conceptual figure of YR main stream, gauging stations, and artificial reservoirs. The sizes of circles indicate the regulation capacities of these reservoirs (Table 2).

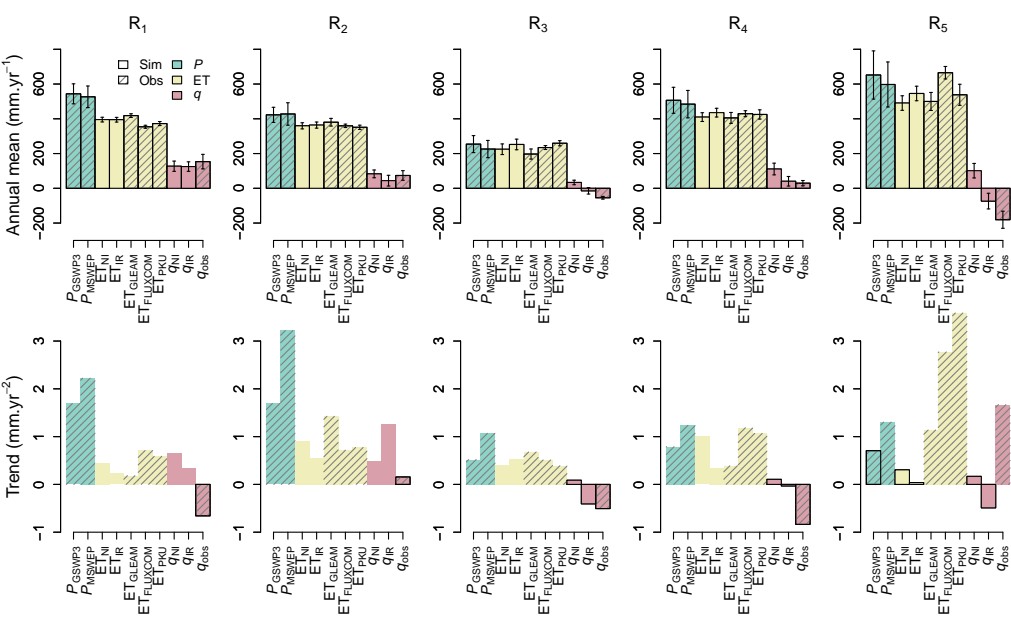

**Figure 2.** Top panel: Annual mean of hydrological elements in each sub-catchment of the YR basin from both simulation (plain colors) and observation (hatched colors). Error bars represent for standard deviation. Bottom panel: trends of these elements in each sub-catchment. Dark borders indicate the trend is statistical significant ($p$-value $< 0.05$) according to Mann-Kendall test.

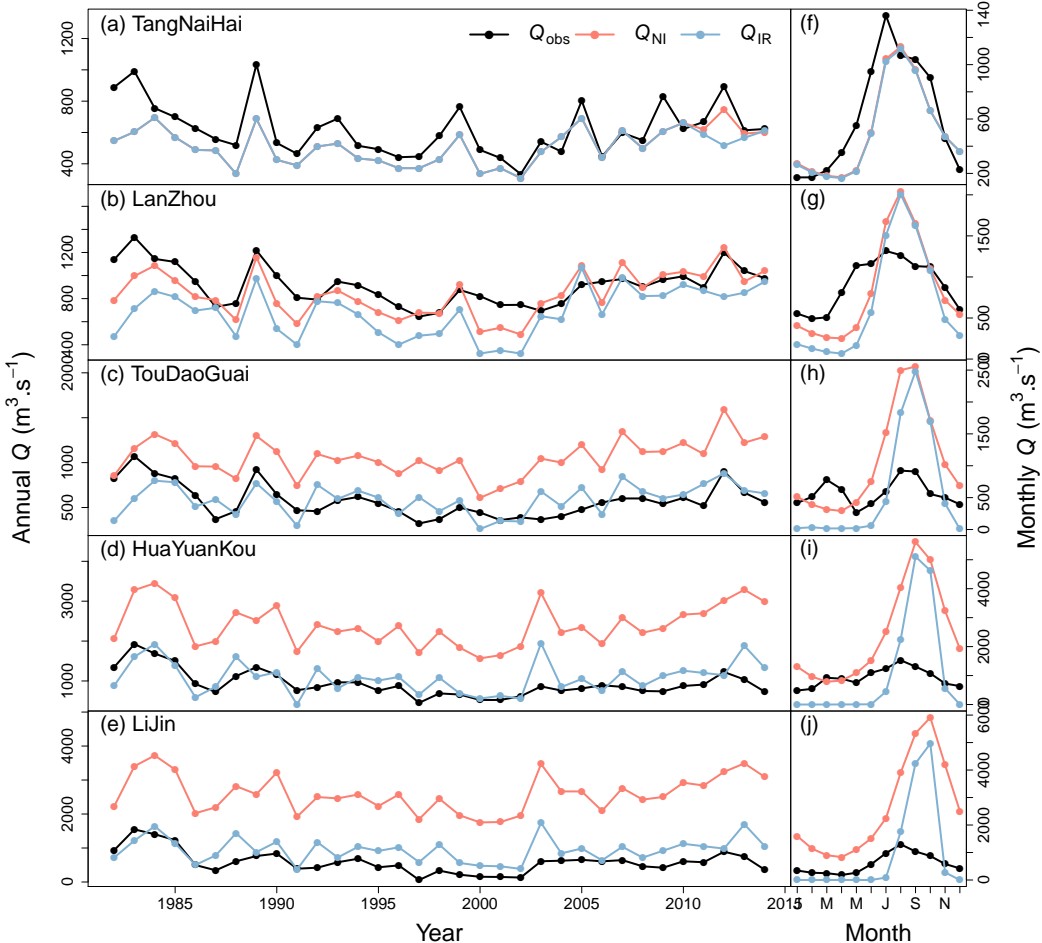

**Figure 3.** (a)-(e): Time series of annual streamflows from observations and simulations at each gauging station. (f)-(j): Seasonality of observed and simulated streamflows at each gauging station. $Q_{obs}$ is the observed annual mean streamflows. $Q_{NI}$ and $Q_{IR}$ are the simulated annual mean streamflows based on the NI and IR simulations (Sect. 2.4), respectively. These simulations do not account for dams and therefore the seasonality has a higher amplitude than observed in the right hand plots.

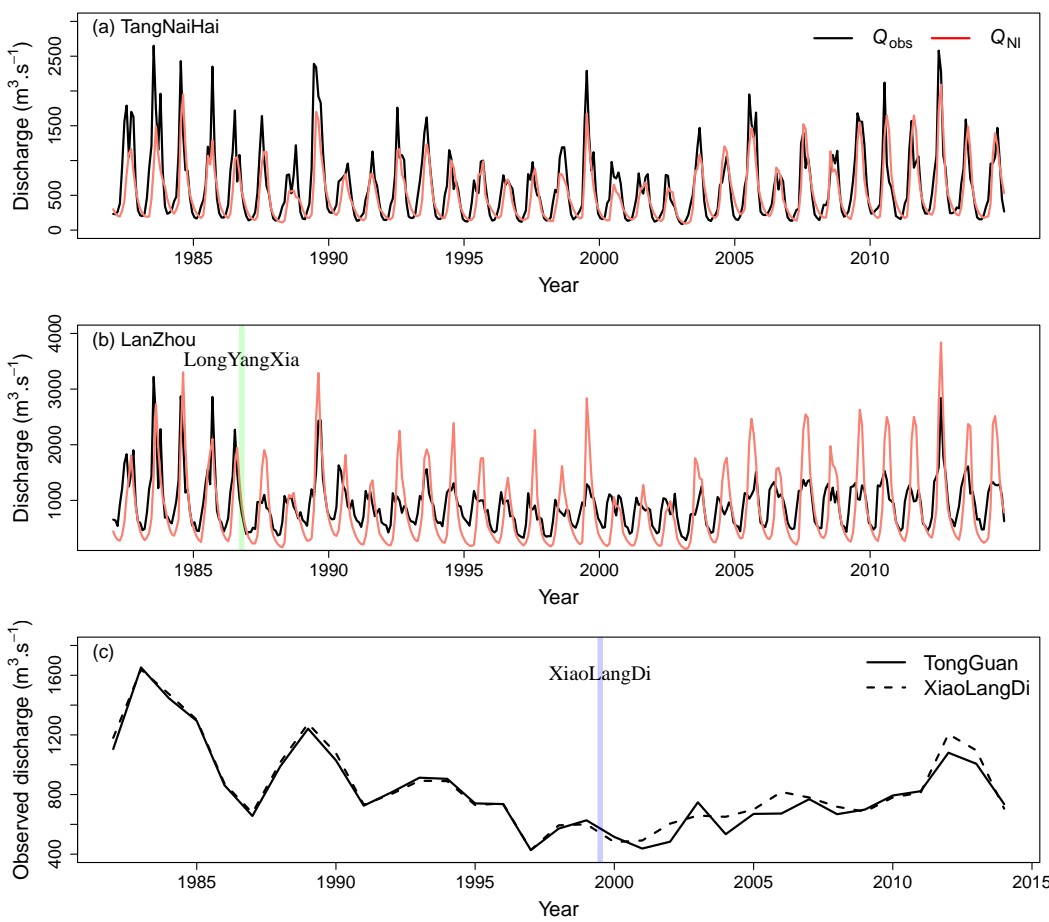

**Figure 4.** (a)-(b): monthly observed ($Q_{obs}$) and simulated ($Q_{NI}$) streamflows at TangNaiHai and LanZhou stations. Green bar in (b) indicates the start of the LongYangXia dam regulation. (c): Observed annual streamflows at TongGuan and XiaoLangDi gauging stations, which locate at up and down stream of the XiaoLangDi reservoir, respectively (see Fig. 1). Blue bar in (c) indicates the start of the XiaoLangDi dam regulation.

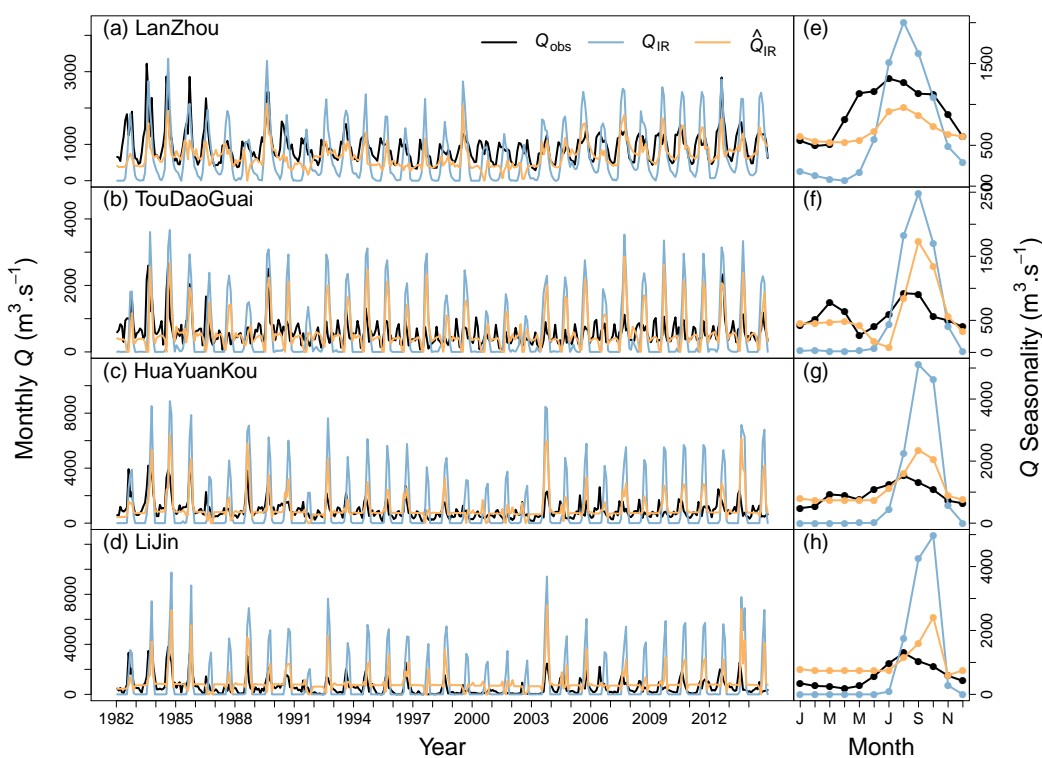

**Figure 5.** Comparison between observed and simulated actual monthly streamflows at gauging stations. $Q_{obs}$ (dark lines) is observed monthly streamflows. $Q_{IR}$ (blue lines) is simulated monthly streamflows from the IR experiment (Sect. 2.4). $\hat{Q}_{IR}$ (orange lines) is simulated monthly streamflows including impacts of dam regulation (Sect. 2.4).

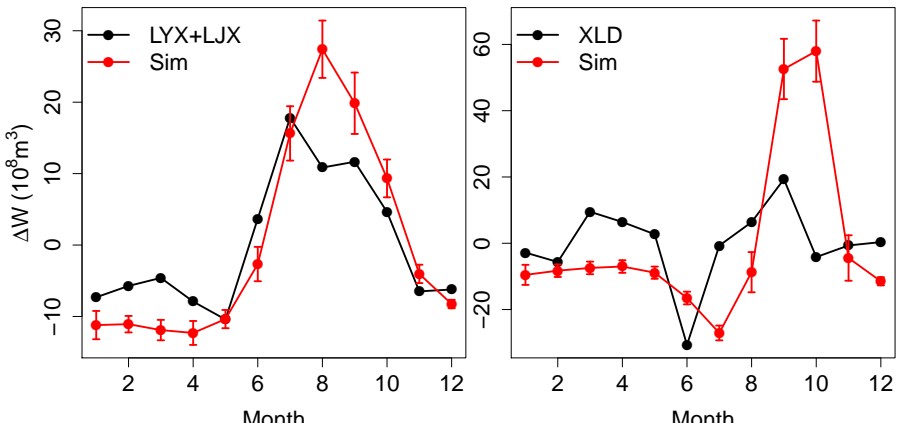

**Figure 6.** The changes of water storage of dams ($\Delta W$) in $R_2$ and $R_4$. The dark line represents the $\Delta W$ from literature. The multi-year mean of $\Delta W$ of LongYangXi and LiuJiaXia is from Jin et al. (2017). The $\Delta W$ of XiaoLangDi is from one-year record reported in Kong et al. (2017). Red lines represent corresponding simulated $\Delta W$ from our dam regulation model.

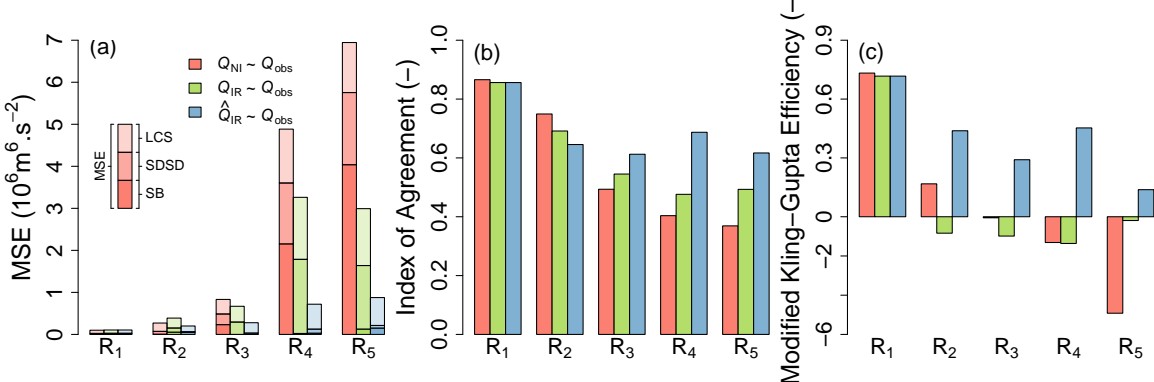

**Figure 7.** Indicators of $Q$ comparisons in each sub-catchment of YRB. Colors indicate different comparisons. The MSE is decomposed to SB, SDSD, and LCS, which are distinguished by different transparencies.