# Peer review of "Irrigation, damming, and streamflow fluctuations of the Yellow River"

_Hydrology and Earth System Sciences, 2020_

## Referee Comment (RC1) · Anonymous Referee #1 · 23 May 2020

This paper presents a modeling study of the effects of irrigation and dams on streamflow changes in the Yellow River Basin. There are many similar attribution studies in the literature looking at various influencing factors in the study region. Authors argue that streamflow fluctuations are not well examined in previous studies. But I am not convinced that this attempt would lead to a significant advance in this field. Besides, there are several issues with the model setup and experiments, which require significant improvement and clarifications to enhance the robustness of this study. Below are my major concerns that should be addressed.

1. The main drawback of this modeling study lies in the coarse resolution of the simulations. The hydrological modeling community has advanced significantly towards hypo-resolution simulations, especially at the river basin scale. Here, authors conduct

[Figure]

the simulations at a spatial resolution of 0.5ox0.5o in the river basin, using global-scale products for model inputs and validations. I believe authors should utilize local data for configuring their model in this specific river basin, given the availability of various high-resolution meteorological forcing data in China and ET products as well.

2. Extensive calibrations should be performed before using the model for quantifying the anthropogenic impacts. Authors argue that streamflow fluctuations have not been well examined in previous studies. but in figure 5-6, the model shows rather poor performance in simulating the seasonality and the peak streamflow, even with consideration of irrigation and dams.

3. In the irrigation scheme, irrigation water requirement is met only by the available stream water. How is the water availability defined? How does the model perform in simulating irrigation water use, compared to census data?

4. In the abstract, "Irrigation is found to be the dominant factor leading to 63.7% reduction of the annual discharges", Is streamflow reduction caused by anthropogenic factors only? How about the effects of changing climate? Authors need to show the relative contribution of each factor (including irrigation) to streamflow changes in the abstract and conclusion sections.

---

## Referee Comment (RC2) · Anonymous Referee #2 · 24 Jun 2020

The study "Irrigation, damming, and streamflow fluctuations of the Yellow River" by Yin et al. provides an overview of the water budget in the Yellow River basin, by considering irrigation and dam regulations. In this study, the authors developed a simple dam model coupled with ORCHIDEE to represent the major flow regulations in the river basin. The topic fits the scope of HESS, However, as a scientific manuscript, a clearly defined science question is missing in this study. What is your major contribution to the hydrology community as the concept of modeling dam regulation is not new? Also, there are many technical issues need to be addressed and improved (see below) before this paper can be considered for publication.

Page 1, line 5, line 10: new -> newly

Page 4, lines 7-8: Although it's true that many dam model algorithms in recent GHMs

and LSMs are inherited from Hanasaki et al. (2006), it is worth mentioning there are other types of dam/reservoir models such as agent-based models (e.g. Riverwave), or basin-specific models (e.g. USBR Colorado River Simulation System).

Page 4, line 23: Remove "real" before observations. Are there "unreal" observations?

Page 4, lines 29-30: I'm not convinced that the new dam model "does not require any prior information from observation". In my opinion, observed information include the data or parameters measured/collected from the real world. In this case, the location, storage capacity, geometry of the dam and reservoir, etc. They are all "observations". So, I feel this sentence (and the one in the abstract) is a bit overselling the model and needs to be further clarified.

Section 2.1.1: Could you add some more background about ORCHIDEE before introducing ORCHIDEE-CROP? What's the relationship between these two? Is ORCHIDEE-CROP an offline crop model taking ORCHIDEE output as input, or it's an updated ORCHIDEE with an online crop model, or it's a regional model only focuses on China?

Section 2.1.2: This scheme concept is quite similar to Voisin et al. (2013). Considering citing the work.

Section 2.1.2: Essentially the dam model is trying to flatten the hydrograph. Any support from the observation that all dams follow this generic rule? I understand sometimes it's hard to obtain the actual operation rules from the dam operators, but given this is a basin scale analysis (not global), some level of "fact-checking" needs to be included to reflect the local reality.

Page 8, line 22: Since NI and IR are major simulation experiments performed in this study, it is necessary to include more descriptions about the irrigation scheme in Section 2.1.1. For example, how does the irrigation demand be evaluated, at what time step? How does the irrigation water be applied, at what time step? I'm assuming

different PFTs are associated with different irrigation methods (e.g. drip, sprinkler, or flood)? How does the return flow be treated in the model? How does the groundwater be represented in the model? If no groundwater pumping is represented in the model, the level of uncertainty needs to be evaluated and discussed for the study basin.

Page 10, line 5: I don't understand why ETni and ETir had no significant differences as I can see the discharge had significant decreases at some gauges (Figure 3). I assume the reduced Q is due to the irrigation water withdrawal, and then become additional ET through the irrigation, or it's not the case here?

Page 10, line 9: In this equation, Ai is the total drainage area between two gauges. Will it make more sense to use irrigated area instead of total area? This way you can compare the relative level of irrigation for different sub-regions?

Page 11, line 16: There are many negative spikes in Q_hat_IR time series in Figure 5. This is unacceptable. I don't think your model is doing the right thing.

Figure7: Given it's a regional study, I'm expecting better results than this, especially when you mentioned some previous study reached NSE around 0.9 for natural flow in the very same basin. Theoretically speaking, the inclusion of irrigation and dam regulation would improve the performance, not the opposite. I think more discussion about this issue is required. Also, how confident are you about the numbers in the conclusion?

Figure 7: NSE is good for evaluating high frequency flow data but might not be a good metric for monthly time series, as it is more sensitive to the peak values (Krause et al. 2005). Maybe that's why your NSE is so bad. I would suggest removing this metric.

Voisin N, Li H, Ward D, Huang M, Wigmosta M, Leung LR. On an improved sub-regional water resources management representation for integration into earth system models. Hydrol Earth Syst Sci 2013;17:3605e22. http:// dx.doi.org/10.5194/hess-17-3605-2013.

Krause, P., Boyle, D. P., and Bäse, F.: Comparison of different efficiency criteria for hydrological model assessment, Adv. Geosci., 5, 89–97, https://doi.org/10.5194/adgeo-5-89-2005, 2005.

---

## Author Comment (AC1) · 6 Aug 2020

The comment was uploaded in the form of a supplement:
https://hess.copernicus.org/preprints/hess-2020-7/hess-2020-7-AC1-supplement.pdf
* * *

---

## Author Comment (AC2) · 6 Aug 2020

**Reply to Referee #2**

Z. Yin on behalf of all co-authors

**1 "The study 'Irrigation, damming, and streamflow fluctuations of the Yellow River' by Yin et al. provides an overview of the water budget in the Yellow River basin, by considering irrigation and dam regulations. In this study, the authors developed a simple dam model coupled with ORCHIDEE to represent the major flow regulations in the river basin. The topic fits the scope of HESS, However, as a scientific manuscript, a clearly defined science question is missing in this study. What is your major contribution to the hydrology community as the concept of modeling dam regulation is not new?"**
A: Thank you very much for your comments. There are two objectives of this study. First, with newly developed crop and irrigation module, the land surface model OR-CHIDEE must be evaluated whether it is able to simulate the discharge of complex rivers with a generic parameterization and to explain the mismatch of simulated discharge of the Yellow River in our previous study (Xi et al., 2018). Moreover, the dam operation model should be evaluated before integrated into ORCHIDEE.
Second, we aim to quantify the impacts of irrigation and dam operations on the monthly discharge fluctuations of the Yellow River, which is not well demonstrated in previous studies. In the revised manuscript, we underlined, "This study aims to 1) demonstrate whether the global land surface model ORCHIDEE is able to simulate the streamflows of complex rivers with human activities using a generic parameterization, and 2) quantify the respective roles of irrigation and artificial reservoirs in monthly streamflow fluctuations of the Yellow River from 1982 to 2014 by using ORCHIDEE with a newly developed irrigation module, and an offline dam operation model." In comparison to previous studies, there are several advantages in our work. Details are discussed in our reply to comment 1 of Referee #1.

**2 "Page 1, line 5, line 10: new → newly"**
A: Corrected.

**3 "Page 4, lines 7-8: Although it's true that many dam model algorithms in recent GHMs and LSMs are inherited from Hanasaki et al. (2006), it is worth mentioning there are other types of dam/reservoir models such as agent-based models (e.g. Riverwave), or basin-specific models (e.g. USBR Colorado River Simulation System)."**
A: Thanks. We've added them in the short review of dam model development.

**4 "Page 4, line 23: Remove 'real' before observations. Are there 'unreal' observations?"**

A: Sorry for the confusion. It has been removed.

**5 "Page 4, lines 29-30: I'm not convinced that the new dam model 'does not require any prior information from observation'. In my opinion, observed information include the data or parameters measured/collected from the real world. In this case, the location, storage capacity, geometry of the dam and reservoir, etc. They are all 'observations'. So, I feel this sentence (and the one in the abstract) is a bit overselling the model and needs to be further clarified."**

A: True. The dam model does require information like regulation capacity, location, and the year when regulation started. This part has been removed in the revision.

**6 "Section 2.1.1: Could you add some more background about ORCHIDEE before introducing ORCHIDEE-CROP? What's the relationship between these two? Is ORCHIDEE-CROP an offline crop model taking ORCHIDEE output as input, or it's an updated ORCHIDEE with an online crop model, or it's a regional model only focuses on China?"**

A: ORCHIDEE-CROP is a special branch of ORCHIDEE with an online crop model, which will be merged with the trunk version after extensive evaluation. It has been applied widely in current research. To avoid this confusion, we removed ORCHIDEE-CROP in the revision. A short introduction of ORCHIDEE and this special version has been added in the revision as: "ORCHIDEE is a physical process-based land surface model that integrates hydrological cycle, surface energy balances, carbon cycle, and vegetation dynamics by two main modules. The SECHIBA (surface-vegetation-atmosphere transfer scheme) module simulates the dynamics of water cycle, energy fluxes, and photosynthesis at half-hourly time interval, which are used by the STOMATE (Saclay Toulouse Orsay Model for the Analysis of Terrestrial Ecosystems) to estimate vegetation and soil carbon cycle at daily time step. The ORCHIDEE used in this study is a special version with newly developed crop and irrigation module (Wang et al., 2017; Wu et al., 2016; Yin et al., 2020). The novel crop module includes specific parameterizations for three main staple crops: wheat, maize, and rice, which are calibrated over China by observations (Wang, 2016; Wang et al., 2017). It is able to simulate crop carbon allocation, different phenological stages as well as related managements (e.g., planting date, rotation, multi-cropping, irrigation, etc)."

**7 "Section 2.1.2: This scheme concept is quite similar to Voisin et al. (2013). Considering citing the work."**

A: Thanks. It has been cited in the introduction of the dam model framework.

**8 "Section 2.1.2: Essentially the dam model is trying to flatten the hydrograph. Any support from the observation that all dams follow this generic rule? I understand sometimes it's hard to obtain the actual operation rules from the dam operators, but given this is a basin scale analysis (not global),**

**some level of 'fact-checking' needs to be included to reflect the local reality."**
A: The functions of main artificial reservoirs in the YRB has been collected from the Yellow River Conservancy Commission of the Ministry of Water Resources (`http://www.yrcc.gov.cn/hhyl/sngc/`), and has been added in Table 1 in the revised manuscript. The information confirms that flood control ('C' in Table 1), irrigation ('I'), and water supply ('W') are primary targets of these reservoirs, which, in principle, would flatten the hydrograph (seems impossible to release water for water supply and irrigation during flooding season, or reduce the discharge during the dry season).

9 **"Page 8, line 22: Since NI and IR are major simulation experiments performed in this study, it is necessary to include more descriptions about the irrigation scheme in Section 2.1.1. For example, how does the irrigation demand be evaluated, at what time step? How does the irrigation water be applied, at what time step? I'm assuming different PFTs are associated with different irrigation methods (e.g. drip, sprinkler, or flood)? How does the return flow be treated in the model? How does the groundwater be represented in the model? If no groundwater pumping is represented in the model, the level of uncertainty needs to be evaluated and discussed for the study basin."**
A: We've improved the introductions of the irrigation module in Section 2.1.1 as: "The water resources in ORCHIDEE account for three water reservoirs: 1) the stream reservoir indicates streamflows; 2) the fast reservoir indicates surface runoff; and 3) the slow reservoir indicates total deep drainage, the order of which indicates the priorities of water reservoirs considered for irrigation. As long-distance water transfer is not taken into account, streams only supply water to the crops growing in the grid-cell they cross, according to the river routing scheme of the ORCHIDEE model (Ngo-Duc et al., 2007)."
and the simulation protocol in Section 2.4 as: "In IR, only surface irrigation is considered in this study (irrigated water is applied on the cropland surface without interception by canopies), which only works during the crop growth period. The soil water stress, a function of profiles of soil moisture and crop root density (up to 2 m depth, (Yin et al., 2020)), is checked every half an hour. When it is less than a target threshold (=1), irrigation will be triggered with amount equal to the deficit of saturated and current soil moisture. To precisely estimate irrigation water consumption (direct water loss from the surface water pool excluding return flow), the deep drainage of the three crop soil columns is turned off in the IR simulation."
The irrigation demand is checked every half an hour. If water stress excesses predefined threshold, irrigation will be triggered. Due to lack of information about irrigation techniques for specific crops, only surface irrigation is applied. If irrigated rate is larger than the infiltration rate, surface runoff will occur, which however is almost forbidden by constraining the irrigation rate. To give a precisely estimation of irrigation consumption, the deep drainage of crop soil columns is turned off. Therefore, the irrigated water can only be used for evapotranspiration. Note that soil water in natural vegetation soil columns still can be lost by deep drainage, which forms the slow reservoir (shallow ground water) that can be withdrawn for irrigation as well. The fossil ground water pumping is not

taken into account in our model. Firstly, the interactive mechanisms between shallow and fossil ground water is now well known (Scanlon et al., 2018). Secondly, there is rare data about the accessibility of deep fossil ground water. Nevertheless, in our previous study (Yin et al., 2020), by using ORCHIDEE-estimated irrigation water withdrawal and a proportion of surface water withdrawal versus ground water withdrawal derived from census data, we successfully explained the trend of total water storage in the YRB (simulated trend is -5.4 mm.yr$^{-1}$; GRACE based trend is -5.36 mm.yr$^{-1}$).

**10 "Page 10, line 5: I don't understand why $ET_{NI}$ and $ET_{IR}$ had no significant differences as I can see the discharge had significant decreases at some gauges (Figure 3). I assume the reduced $Q$ is due to the irrigation water withdrawal, and then become additional ET through the irrigation, or it's not the case here?"**

A: Here we compared the magnitudes of simulated ET and observed (or satellite-based) ET, the differences between which is not significant (differences are smaller than the variation of observed ET among different products). In fact, simulated ET coincides well with the observations (Table S1). True. The $ET_{IR}$ is always higher than $ET_{NI}$ due to the irrigation withdrawal, which also results in $Q_{IR} < Q_{NI}$.

**11 "Page 10, line 9: In this equation, $A_i$ is the total drainage area between two gauges. Will it make more sense to use irrigated area instead of total area? This way you can compare the relative level of irrigation for different sub-regions?"**

A: Thanks for your suggestion. The equation here corresponds to the Equation 8. Here we provided sub-section-based water balance diagnosis. Although it is a good idea to show irrigation intensity (by changing $A_i$ to irrigated area), we should consider the water balances in sub-sections, where precipitation and evapotranspiration – that are not only occur on irrigated area – are taken into account as well. The spatial distribution of irrigation intensity has been illustrated in our previous study (Yin et al., 2020).

**12 "Page 11, line 16: There are many negative spikes in $\hat{Q}_{IR}$ time series in Figure 5. This is unacceptable. I don't think your model is doing the right thing."**

A: Many thanks for your comment which allows us to find and correct an issue in our dam modelling. Indeed, the water recharge of reservoirs was not constrained by inflows and that explains the negative spikes in $\hat{Q}_{IR}$ time series. In the revision, we corrected corresponding equations (Eq. 6) and re-performed the simulations and results.

**13 "Figure7: Given it's a regional study, I'm expecting better results than this, especially when you mentioned some previous study reached NSE around 0.9 for natural flow in the very same basin. Theoretically speaking, the inclusion of irrigation and dam regulation would improve the performance, not the opposite. I think more discussion about this issue is required. Also, how confident are you about the numbers in the conclusion?"**

A: The inclusion of irrigation and dam regulation would dramatically reduce the RMSE,

which has been shown in our result (MSE=RMSE$^2$, Fig. 7a). However, it probably will not lead to a higher NSE of regulated discharge than NSE of naturalized discharge. Here is a simple proof.

Assuming that $N_i$ is the time series of natural discharge and $\Delta W_i$ is water storage change of a reservoir. Thus, the regulated discharge $R_i$ can be calculated as:

$$
\begin{aligned}
R_i &= N_i - \Delta W_i, \\
r_i &= n_i - \Delta w_i.
\end{aligned}
\tag{1}
$$

Where $i$ is month index. Capital letters indicate observed variables; while lower case letters indicate simulated variables. Then the NSE of regulated discharge (NSE$_1$) can be calculated as:

$$
\begin{aligned}
\mathrm{NSE}_1 &= 1 - \frac{\sum\limits_{i=1}^{M} \left(R_i - r_i\right)^2}{\sum\limits_{i=1}^{M} \left(R_i - \bar{R}\right)^2} \\
&= 1 - \frac{\sum\limits_{i=1}^{M} \left[(N_i - \Delta W_i) - (n_i - \Delta w_i)\right]^2}{\sum\limits_{i=1}^{M} \left(R_i - \bar{R}\right)^2},
\end{aligned}
\tag{2}
$$

where $M$ is the length of the time series. Let's assume that the model can give a perfect simulation of water storage change of reservoir. Thus $\Delta w_i = \Delta W_i$ and NSE$_1$ is,

$$
\mathrm{NSE}_1 = 1 - \frac{\sum\limits_{i=1}^{M} \left(N_i - n_i\right)^2}{\sum\limits_{i=1}^{M} \left(R_i - \bar{R}\right)^2}.
\tag{3}
$$

Note that the NSE of natural discharge (NSE$_2$) is,

$$
\mathrm{NSE}_2 = 1 - \frac{\sum\limits_{i=1}^{M} \left(N_i - n_i\right)^2}{\sum\limits_{i=1}^{M} \left(N_i - \bar{N}\right)^2}.
\tag{4}
$$

The difference between NSE$_1$ and NSE$_2$ is the variation of regulated and natural discharge. As assuming that dam operations always reduce the variation of discharge, the variation of $N_i$ is smaller than $R_i$. Consequently, NSE$_2$ is always less than NSE$_1$. In summary, if reservoirs reduce the variation of river discharge, a model even with **a perfect dam module** will always provide a smaller NSE (with regulated discharge as reference) than that of the model without functions of dam operations (with natural discharge as reference)! The conclusion is that it is not comparable of model (study) performances

with different references and that it is not adequate to evaluate dam parameterizations. This proof has been added in the online supplement. And in Sect. 4 we discussed: "These NSE decreases were interpreted due to the complexity of the YRB under the impacts of human activities and climate variation. However, the NSE of natural discharges is incomparable to the NSE of regulated discharges. Even if the model can perfectly simulate the reservoir operations, the NSE of natural discharges is certainly larger than that of regulated discharges from the same model, if you accept the assumption that reservoir operations reduce the variation of river streamflows (a simple proof is available in Sect. A in the online supplement)."

14 **"Figure 7: NSE is good for evaluating high frequency flow data but might not be a good metric for monthly time series, as it is more sensitive to the peak values (Krause et al. 2005). Maybe that's why your NSE is so bad. I would suggest removing this metric."**
A: True. NSE is more sensitive to peak flows than base flows. It is ideal for short-term flood prediction. However, for studies concentrating the resilience of human society to water resources variation, how much base discharge that reservoirs are able to guarantee will be more interesting, in the case of which NSE probably is not suitable. Moreover, we recognize that it is unfair to compare NSEs of natural discharge to that of regulated discharge (see our reply to Comment 13). In short, we agree with your suggestion and removed the NSE in the revised manuscript. The evaluation is now performed using the complementary criteria: KEG, MSE and index of agreement.

**Bibliography**

Ngo-Duc, T., Laval, K., Ramillien, G., Polcher, J., and Cazenave, A.: Validation of the land water storage simulated by Organising Carbon and Hydrology in Dynamic Ecosystems (ORCHIDEE) with Gravity Recovery and Climate Experiment (GRACE) data, Water Resources Research, 43, 1–8, https://doi.org/10.1029/2006WR004941, 2007.

Scanlon, B. R., Zhang, Z. Z., Save, H., Sun, A. Y., Müller Schmied, H., van Beek, L. P. H., Wiese, D. N., Wada, Y., Long, D., Reedy, R. C., Longuevergne, L., Döll, P., and Bierkens, M. F. P.: Global models underestimate large decadal declining and rising water storage trends relative to GRACE satellite data, Proceedings of the National Academy of Sciences, 115, E1080–E1089, https://doi.org/10.1073/pnas.1704665115, URL http://www.pnas.org/lookup/doi/10.1073/pnas.1704665115, 2018.

Wang, X. H.: Impacts of environmental change on rice ecosystems in China: development, optimization and application of ORCHIDEE-CROP model, Ph.D. thesis, Peking University, 2016.

Wang, X. H., Ciais, P., Li, L., Ruget, F., Vuichard, N., Viovy, N., Zhou, F., Chang, J. F., Wu, X. C., Zhao, H. F., and Piao, S. L.: Management outweighs climate change on affecting length of rice growing period for early rice and single rice in China during 1991–2012, Agricultural and Forest Meteorology, 233, 1–11, https://doi.org/10.1016/j.agrformet.2016.10.016, URL http://linkinghub.elsevier.com/retrieve/pii/S0168192316304087, 2017.

Wu, X. C., Vuichard, N., Ciais, P., Viovy, N., de Noblet-Ducoudré, N., Wang, X. H., Magliulo, V., Wattenbach, M., Vitale, L., Di Tommasi, P., Moors, E. J., Jans, W., Elbers, J., Ceschia, E., Tallec, T., Bernhofer, C., Grünwald, T., Moureaux, C., Manise, T., Ligne, A., Cellier, P., Loubet, B., Larmanou, E., and Ripoche, D.: ORCHIDEE-CROP (v0), a new process-based agro-land surface model: model description and evaluation over Europe, Geoscientific Model Development, 9, 857–873, https://doi.org/10.5194/gmd-9-857-2016, URL http://www.geosci-model-dev-discuss.net/8/4653/2015/http://www.geosci-model-dev.net/9/857/2016/, 2016.

Xi, Y., Peng, S. S., Ciais, P., Guimberteau, M., Li, Y., Piao, S. L., Wang, X. H., Polcher, J., Yu, J. S., Zhang, X. Z., Zhou, F., Bo, Y., Ottle, C., and Yin, Z.: Contributions of Climate Change, $CO_2$, Land-Use Change, and Human Activities to Changes in River Flow across 10 Chinese Basins, Journal of Hydrometeorology, 19, 1899–1914, https://doi.org/10.1175/JHM-D-18-0005.1, URL http://journals.ametsoc.org/doi/10.1175/JHM-D-18-0005.1, 2018.

Yin, Z., Wang, X. H., Ottlé, C., Zhou, F., Guimberteau, M., Polcher, J., Peng, S. S., Piao, S. L., Li, L., Bo, Y., Chen, X. L., Zhou, X. D., Kim, H., and Ciais, P.: Improvement of the Irrigation Scheme in the ORCHIDEE Land Surface Model and Impacts of Irrigation on Regional Water Budgets Over China, Journal of Advances

in Modeling Earth Systems, 12, 1–20, https://doi.org/10.1029/2019MS001770, URL https://onlinelibrary.wiley.com/doi/abs/10.1029/2019MS001770, 2020.

---

## Author Response (AR1)

**Reply to Referee #1**

Z. Yin on behalf of all co-authors

1 "This paper presents a modeling study of the effects of irrigation and dams on streamflow changes in the Yellow River Basin. There are many similar attribution studies in the literature looking at various influencing factors in the study region. Authors argue that streamflow fluctuations are not well examined in previous studies. But I am not convinced that this attempt would lead to a significant advance in this field."

A: Thank you very much for your comments. It is true that many attribution studies have been performed in the Yellow River Basin (YRB). But different from them, there are three main advantages is this study.

First, novel crop module and China's Plant Functional Types (PFT) map were used in this work. Accurate crop simulation is a precondition of reasonable irrigation estimation. Some previous studies do not have crop simulations and need observed or satellite-based data (e.g., Leaf Areas Index and fraction of photosynthetically active radiation absorbed by green vegetation) to drive their irrigation simulations. Although some Global Hydrological Models and Global Land Surface Models (GHMs and GLSMs) did develop their crop modules, the crop functions, which are always based on C3 grass generics parameterizations, are too coarse to simulation varied crop types and phenology over China. The lack of physical-processes based crop dynamic simulations of previous studies has been discussed (Page 4, Line 72–79) as:" Many model studies are able to provide reliable estimation of river discharges but related physical processes are not fully represented. For instance, some model studies require extra observed data as inputs (e.g., leaf area index (LAI), evapotranspiration, etc). Moreover, many biophysical processes (e.g., photosynthesis, LAI dynamics, crop phenology), which tightly couple with evapotranspiration, surface energy balances, and irrigation demands, are rarely considered in Global Hydrological Models (GHM). These missing processes are not important for hydrological studies using historical data and short-term forecast. However, they are probably non-negligible for long-term projections (Duethmann et al., 2020), especially in regions where ecosystems react strongly to climate change through the hydrological cycle (de Boer et al., 2012; Lian et al., 2020; Zhu et al., 2016)." The novel crop module in ORCHIDEE is able to simulate most physical processes throughout the whole crop growth period (Wang, 2016), It has specific parameterizations for wheat, maize and rice, which are the three main staple crops in China, which have been calibrated based on census data (Wang et al., 2017). The advantages of the novel crop module of ORCHIDEE has been introduced in the manuscript (Page 4, Line 84–86) as: "By developing a new

crop-irrigation module in ORCHIDEE (Wang et al., 2016; Wang, 2016; Wu et al., 2016; Yin et al., 2020), we were able to provide precise estimation of crop phenology, yield and irrigation amount at both local and national scales (Wang et al., 2017; Yin et al., 2020)." Moreover, the novel China's PFT map has been developed including the fractions of wheat, maize, and rice based on 1:1 million vegetation map and provincial scale census data from the National Bureau of Statistics. For the first time, the irrigation consumption is estimated based on varied phenology of different crop types in different regions. The introduction of the performances of ORCHIDEE-simulated irrigation as been added (Page 4, Line 86 – Page 5, Line 90) as:"More importantly, ORCHIDEEestimated irrigation accounts for potential ecological and hydrological impacts (e.g., physiological response of plants to climate change and short term drought episodes on soil hydrology) with respect to other land surface models and global hydrological models. In a study focusing on China (Yin et al., 2020), ORCHIDEE estimated irrigation withdrawal coincided well with census data (provincial-based spatial correlations are  $^{\circ}0.68$ ), and successfully explained the decline of total water storage in the YRB." And the novel China's PFT map has been revised (Page 9, Line 210–214) as: "A 0.5° map with 15 different Plant Functional Types (PFTs) containing crop sowing area information for the three PFTs corresponding to the modeled crop (wheat, maize, and rice) is used, based on 1:1 million vegetation map and provincial scale census data of China. Crop planting dates for wheat, maize, and rice are derived from spatial interpolation of phenological observations from Chinese Meteorological Administration (Wang et al., 2017)." Second, we simulate river discharges and dam operations in the YRB and validate them on a recent time period. Some global studies simulated the Yellow River with irrigation and dam operations. But the period of most simulations starts from 1960s or 1970s,

when a high proportion of discharges was less affected by dams. In this study, we focus on the period when huge reservoirs (LongYangXia in 1986 and XiaoLangDi in 1999) started regulation. We underlined this point in the revision (Page 4, Line 64–71) as: "Although large uncertainties among model simulations are addressed (Haddeland et al., 2014: Liu et al., 2019), rare studies focus on the YRB to demonstrate where the errors of simulations from due to lack of data. Moreover, the validation periods of many modelling studies started from 1960~1970 to 2000~2010 (Haddeland et al., 2014; Hanasaki et al., 2018; Liu et al., 2019; Tang et al., 2008; Wada et al., 2016) whereas several large reservoirs started regulation much later (e.g., LongYangXia in 1986 and XiaoLangDi in 1999). Such high proportions of observed streamflows rarely affected by reservoirs (>40%) probably cannot guarantee the abilities of models in simulating reservoir operations being correctly evaluated. Thus it is crucial to zoom in the reservoir-dominated period of the YRB to demonstrate the impacts of reservoirs on flow fluctuations under validation by observed dam operations." More importantly, we are the first to show the simulated water storage change of reservoirs and to validate it with observations from literature. The correlation coefficient of simulated and observed water storage change of LiuJiaXia and LongYangXia is over 0.9 (Fig. 6), suggesting that the dam model is able to reproduce dam operations under climate variations. This achievement has been highlighted in the abstract (Page 1, Line 12–15) as: "Inclusion of dam operation dramatically reduced the MSE of simulated discharge by  $\geq 48.4\%$  compared to the simulation only considering irrigation, and increased the predictability of water storage changes of the LongYangXia and LiuJiaXia reservoirs (correlation coefficient of ~0.9)."

Third, detailed diagnosis of anthropogenic factors in the YRB. Many global studies admit the complexity in simulating the streamflows of the YRB (Haddeland et al., 2014; Hanasaki et al., 2018; Wada et al., 2014, 2016). However, rare studies demonstrate where the mismatches from, and whether any key factor or mechanism is missing in the model. Through reviewing literature and reports, we demonstrated several possible important factors (mechanisms) missed in current simulations in the YRB, which are not well represented in GHMs and GLSMs as well. Details are discussed in our reply to Comment 3.

2 "1. The main drawback of this modeling study lies in the coarse resolution of the simulations. The hydrological modeling community has advanced significantly towards hypo-resolution simulations, especially at the river basin scale. Here, authors conduct the simulations at a spatial resolution of  $0.5^{\circ} \times 0.5^{\circ}$  in the river basin, using global-scale products for model inputs and validations. I believe authors should utilize local data for configuring their model in this specific river basin, given the availability of various high-resolution meteorological forcing data in China and ET products as well."

A: To pursue accurate river discharge simulations, many hydrological models used high resolution atmospheric forcing (like 10 km) as driver. However, different from their objective for short-term flood prediction, our aim is to understand the mechanisms and discover missing mechanisms of how human activities affect the discharge fluctuations in the YRB, for which high resolution forcing is not necessary. In fact, our previous study (Xi et al., 2018) utilized 0.1° forcing (Chen et al., 2011) to attribute different factors to the trends of streamflows over China, which showed large overestimation of Yellow River annual discharge. Thus, the crucial questions, which are our objectives as well, are whether irrigation can explain the discharge overestimation in Xi et al. (2018) and what is the impact of dam operations on the river streamflow. Obviously, increasing spatial resolution is not helpful to interpret the mismatch. We agree with the referee's comments that high-resolution forcing is compulsory for accurate simulations. But before that, all important mechanisms should be implemented in the model. In the revised introduction, we introduced our previous study using high resolution forcings and emphasised that the crucial problem is mechanism missing (Page 4, Line 80–84) as: "In our previous study, Xi et al. (2018) utilized 0.1° hypo-resolution atmospheric forcing of China (Chen et al., 2011) to drive the land surface model ORCHIDEE (ORganizing Carbon and Hydrology in Dynamic EcosystEms) in aim to attribute the trends of main China's river streamflows to several natural and anthropogenic factors. Due to lack of representation of crop and irrigation processes, simulated results are consistent to the naturalized streamflows of the YR, however much higher than the observations..."

In fact, the GSWP3 forcing has been corrected by a suite of ground-based observations (http://hydro.iis.u-tokyo.ac.jp/GSWP3/exp1.html#boundary-conditions). For instance, its precipitation assimilates with the GPCC (Global Climatology Centre) precipitation dataset that includes numerous gauges intensively distributed over China (Fig. R1, Becker et al. (2013)). Long-term (1982–2014) in-situ ET measurements (eddy covariance) that are still rare over China, particularly in the YRB (Chen et al., 2014; Lian et al., 2018). Although uncertainties exist in global ET products, they are able to reflect monthly ET magnitude and inter-annual variations (Pan et al., 2020). Nevertheless, our previous study (Yin et al., 2018) validated ORCHIDEE-simulated soil moisture (which indirectly reflects ET dynamics) over China by in-situ measurements, which shows a good agreement (median correlation coefficient 0.53 and RMSE  $0.07 \text{ m}^3.\text{m}^{-3}$ ).

Figure R1: The map of 67,200 gauging stations used for the GPCC precipitation data production (from Becker et al. (2013)).

3 "2. Extensive calibrations should be performed before using the model for quantifying the anthropogenic impacts. Authors argue that streamflow fluctuations have not been well examined in previous studies. but in figure 5-6, the model shows rather poor performance in simulating the seasonality and the peak streamflow, even with consideration of irrigation and dams." A: We agree that model calibration is necessary before utilization for scientific research. Previous studies demonstrate that our model performs well in simulating soil moisture dynamics (Yin et al., 2018), naturalized river streamflows (Table S1 in Xi et al. (2018)), leaf area index (Section S2 in Xi et al. (2018)), amount and trend of irrigation withdrawals (Yin et al., 2020), trends of total water storage (Section 3.4 in Yin et al. (2020)), and ET (Table S1 in online supplement) over China and in the YRB. In the revision we discussed (Page 15, Line 413–418) as:"Although mismatches exist in the simulated discharges, they are unlikely caused by the false representations of physical laws or unsuitable parameterization in our model, because other simulated hydrological variables coincide well with observations in the YRB (e.g., soil moisture dynamics (Yin et al., 2018), naturalized river streamflows (Table S1 in Xi et al. (2018)), leaf area index (Section S2 in Xi et al. (2018)), amount and trend of irrigation withdrawals (Yin et al., 2020), trends of total water storage (Section 3.4 in Yin et al. (2020)), and ET (Table S1))."

However, we cannot fully agree that our model performances are poor in simulating streamflow fluctuations based on Figure 5–6. First, after considering irrigation and dams, the bias of annual discharge and seasonality is substantially reduced (SB and SDSD reduce dramatically in Fig. 7a). Second, our study provides the comparison of simulated and observed water storage change of the LongYangXia and LiuJiaXia reservoirs for the first time. The correlation coefficient is 0.9, which, in our opinion, is quite good given the lack of information of the operation rules. Third, although natural discharge simulations with NSE=0.9 in a small sub-basin of the Yellow River is cited in our study, the NSE of them is incomparable to that of our simulations to conclude that our simulations are poor. A simple proof is given in our reply to the comment 13 from the second referee. In the revised abstract, we underlined that the simulation performances gradually increase with including irrigation and dam operations (Page 1, Line 6–8) as: "Validations with observed discharge near the outlet of the YR demonstrated that model performances improved notably with gradually considering irrigation (mean square error [MSE] decreased 56.9%) and dam regulations (MSE decreased 30.5%) further)."

It is true that mismatches still exist between simulations and observations. However, how to treat these mismatches depends on your goal. If the model services for short- or mid-term streamflow prediction, it is necessary to calibrate the parameters in the model to make the simulated streamflows fit the observations as well as possible regardless the detailed physical processes and other linked variables (e.g., surface energy balances, carbon cycles, vegetation dynamics, etc). However, such approach is probably not conducive to fundamental model improvements in terms of projecting streamflow variations under climate change, because some important missing mechanisms may be obscured by extensive calibrations. For instance, a study highlighted by HESS currently questioned why some well-calibrated models cannot perform well in forecasting river discharges under climate change (Duethmann et al., 2020). Through zooming in to a catchment in Austria, they revealed "the importance of considering interrelations between changes in climate, vegetation and hydrology for hydrological modelling in a transient climate."

On the other hand, which is our case, if the model is used to demonstrate interactive mechanisms among climate, water resources, and human activities, these mismatches should be well investigated rather than be directly calibrated. For instance, we find that our model underestimates the annual discharge at LanZhou in the period 2000–2002 (Fig. 3b), during which  $\hat{Q}_{IR}$  was almost negatively correlated to the  $Q_{obs}$  (Fig. 5a). From China Water Resources Bulletin (2000-2002, http://www.mwr.gov.cn/sj/tjgb/szygb/), we find that to avoid discharge cutoff ( $Q < 1 \text{ m}^3.\text{s}^{-1}$ ) irrigation and hydropower are strictly restricted. It suggests that integrated catchment management plays an important role in river flow variation, especially for extreme years. Obviously, models are not able to reproduce this special reaction by over calibration, if the related mechanisms

are missing.

Moreover, from these mismatches, we also reveal other possible missing factors and mechanisms: 1) the Hetao Plateau withdraws  $50 \times 10^8 \text{ m}^3$  water from the Yellow River, which is neglected in most models because there is no large dam but multiple small reservoirs and complicated channel networks. It may lead to the overestimation of peakflows in Fig. 5; 2) The souring sediment is a special operation target of the XiaoLangDi dam, which release water one month ahead resulting in the delay of simulated water storage change (right panel of Fig. 6). All in all, as the famous statistician George Box said, "All models are wrong, but some are useful" (Box, G. E. P. 1976), if the "wrong" thing in the simulation can help us to discover important missing mechanisms rather than cover them by over calibration, I think the work is "useful". The discussion here are summarized in Sect. 4 of the revised manuscript (Page 14, Line 398 – Page 15, Line 426) as: "... However, when considering the impacts of irrigation and dams, the NSE values of simulations are much worse. For instance, the simulation considering anthropogenic effects from Hanasaki et al. (2018) had lower NSE than the simulation with only natural processes. Similarly, Wada et al. (2014) showed NSE decrease after considering anthropogenic factors in the YRB. These NSE decreases were interpreted due to the complexity of the YRB under the impacts of human activities and climate variation. However, the NSE of naturalized discharges is incomparable to the NSE of regulated discharges. Even if the model can perfectly simulate the reservoir operations, the NSE of naturalized discharges is certainly larger than that of regulated discharges from the same model, if you accept the assumption that reservoir operations reduce the variation of river streamflows (a simple proof is available in Sect. A in the online supplement). In fact, our simulated patterns are very similar with a set of simulations by GHMs (Fig. S2 from Liu et al. (2019)). By gradually considering anthropogenic factors (irrigation and dam operations), the performances of our simulations increase dramatically according to all the three metrics."

"Intensive calibrations or using a suite of observed inputs can allow catchment-scale studies to provide high-accurate simulated discharges for short-term flood forecast. However, the parameterizations are not generic for broad application in other catchments that lack information in particular (Nash and Sutcliffe, 1970). Moreover, insensitive calibrations are not helpful to reveal important mechanisms missed in the model. Without these crucial mechanisms, models hardly to extrapolate their knowledge to predict extreme events and future flood characters under climate change (Duethmann et al., 2020). Unlike them, one aim of our modelling study is to demonstrate interactive mechanisms in a physical-based land surface model. Although mismatches exist in the simulated discharges, they are unlikely caused by the false representations of physical laws or unsuitable parameterization in our model, because other simulated hydrological variables coincide well with observations in the YRB (e.g., soil moisture dynamics (Yin et al., 2018), naturalized river streamflows (Table S1 in Xi et al. (2018)), leaf area index (Section S2 in Xi et al. (2018)), amount and trend of irrigation withdrawals (Yin et al., 2020), trends of total water storage (Section 3.4 in Yin et al. (2020)), and ET (Table S1)). On the contrary, these mismatches draw our attention to some key mechanisms overlooked in most models. For instance, our model underestimates the annual discharge at LanZhou in the period 2000–2002 (Fig. 3b), during which  $\hat{Q}_{IR}$  was almost negatively correlated to the  $Q_{obs}$  (Fig. 5a). From China Water Resources Bulletin (2000–2002, http://www.mwr.gov.cn/sj/tjgb/szygb/), we find that to avoid discharge cut-off ( $Q < 1 \text{ m}^3.\text{s}^{-1}$ ) irrigation and hydropower are strictly restricted throughout the droughts. It suggests that integrated catchment management plays an important role in river flow variation, especially for extreme years. Obviously, models are not able to reproduce this special reaction by over calibration, if the related mechanisms are missing. All in all, mismatches may be useful if they can help us to discover important mechanisms missed before (Duethmann et al., 2020; Scanlon et al., 2018), which is crucial to improve the robustness of a model for future projection."

**4 "3. In the irrigation scheme, irrigation water requirement is met only by the available stream water. How is the water availability defined? How does the model perform in simulating irrigation water use, compared to census data?"**

A: Thanks. It should be "available water resources", which has been corrected in the revised version. The available water resources include three water reservoirs in OR-CHIDEE: 1) stream reservoir (streamflow); 2) fast reservoir (surface runoff); and 3) slow reservoir (deep drainage). Detailed introduction has been added in Section 2.1.1 (Page 6, Line 123–125) as: "The water resources in ORCHIDEE account for three water reservoirs: 1) the stream reservoir indicates streamflows; 2) the fast reservoir indicates surface runoff; and 3) the slow reservoir indicates total deep drainage, the order of which indicates the priorities of water reservoirs considered for irrigation. As long-distance water transfer is not taken into account, streams only supply water to the crops growing in the grid-cell they cross, according to the river routing scheme of the ORCHIDEE model (Ngo-Duc et al., 2007)."

The irrigation module has been introduced and validated in Yin et al. (2020), which shows a good agreement of spatial distribution with census data. In Section 1 (Page 4, Line 88 – Page 5, Line 90) we added: "In a study focusing on China (Yin et al., 2020), ORCHIDEE estimated irrigation withdrawal coincided well with census data (provincial-based spatial correlations are  $\approx 0.68$ ), and successfully explained the decline of total water storage in the YRB."

5 "4. In the abstract, 'Irrigation is found to be the dominant factor leading to 63.7% reduction of the annual discharges'. Is streamflow reduction caused by anthropogenic factors only? How about the effects of changing climate? Authors need to show the relative contribution of each factor (including irrigation) to streamflow changes in the abstract and conclusion sections."

A: As industry and urban water consumptions are not taken into account in this study, we turn to report the amount of irrigation consumption instead of percentage of annual discharge. It is revised (Page 1, Line 9–10) as: "Irrigation is found to substantially reduce the river streamflow by consuming approximately  $242.8 \pm 27.8 \times 10^8 \text{ m}^3.\text{yr}^{-1}$  in line with the census data  $(231.4 \pm 31.6 \times 10^8 \text{ m}^3.\text{yr}^{-1})$ ." The stream reduction here

means the difference between mean annual natural discharge and mean annual observed discharge due to irrigation (call it R1), not the impact of irrigation on the long-term decreasing trend of observed discharge (call it R2, if significant trend exists).

The streamflow reduction (R1) is mainly caused by anthropogenic factors (e.g., water consumption, reservoir surface evaporation, etc). However, the trend of streamflow reduction (R2) is not only caused by anthropogenic factors. Indeed, climate change is the primary driver of trends of the Yellow River streamflows, which has been demonstrated in our previous attribution study including climate change,  $CO_2$  rise, land use change, and human activities (Xi et al., 2018). As this study concentrates on possible impacts of simulating anthropogenic factors on R1, we did not perform the similar analysis shown in Xi et al. (2018). Nevertheless, we demonstrate that climate change, at least the change of precipitation, has little effect on the change of streamflow seasonality (Section. 3.2 and Figure S4). We mentioned this finding in the introduction (Page 1 Line 11–12): "Our analysis revealed that the dam regulation, rather than the change of precipitation, was the primary driver altering streamflow seasonality."

**Bibliography**

- Becker, A., Finger, P., Meyer-Christoffer, A., Rudolf, B., Schamm, K., Schneider, U., and Ziese, M.: A description of the global land-surface precipitation data products of the Global Precipitation Climatology Centre with sample applications including centennial (trend) analysis from 1901-present, Earth System Science Data, 5, 71-99, https://doi.org/10.5194/essd-5-71-2013, URL http://www.earth-syst-sci-data. net/5/71/2013/, 2013.
- Chen, X., Su, Z., Ma, Y., Liu, S., Yu, Q., and Xu, Z.: Development of a 10-year (2001-2010) 0.1° data set of land-surface energy balance for mainland China, Atmospheric Chemistry and Physics, 14, 13097–13117, https://doi.org/ 10.5194/acp-14-13097-2014, URL http://www.atmos-chem-phys.net/14/13097/ 2014/, 2014.
- Chen, Y. Y., Yang, K., He, J., Qin, J., Shi, J. C., Du, J. Y., and He, Q.: Improving land surface temperature modeling for dry land of China, Journal of Geophysical Research, 116, D20104, https://doi.org/10.1029/2011JD015921, URL http://www.tandfonline.com/doi/abs/10.1080/10643380902800034http: //doi.wiley.com/10.1029/2011JD015921, 2011.
- de Boer, H. J., Eppinga, M. B., Wassen, M. J., and Dekker, S. C.: A critical transition in leaf evolution facilitated the Cretaceous angiosperm revolution, Nature Communications, 3, 1221, https://doi.org/10.1038/ncomms2217, URL http://www.pubmedcentral.nih.gov/articlerender.fcgi?artid=3514505{\& }tool=pmcentrez{\&}rendertype=abstracthttp://www.nature.com/doifinder/ 10.1038/ncomms2217, 2012.
- Duethmann, D., Blöschl, G., and Parajka, J.: Why does a conceptual hydrological model fail to correctly predict discharge changes in response to climate change?, Hydrology and Earth System Sciences, 24, 3493-3511, https://doi.org/10. 5194/hess-24-3493-2020, URL https://hess.copernicus.org/articles/24/3493/ 2020/, 2020.
- Haddeland, I., Heinke, J., Biemans, H., Eisner, S., Flörke, M., Hanasaki, N., Konzmann, M., Ludwig, F., Masaki, Y., Schewe, J., Stacke, T., Tessler, Z. D., Wada, Y., and Wisser, D.: Global water resources affected by human interventions and climate change, Proceedings of the National Academy of Sciences, 111, 3251–3256, https://doi.org/10.1073/pnas.1222475110, URL http://www.pnas.org/ lookup/doi/10.1073/pnas.1222475110, 2014.
- Hanasaki, N., Yoshikawa, S., Pokhrel, Y., and Kanae, S.: A global hydrological simulation to specify the sources of water used by humans, Hydrology and Earth System Sciences, 22, 789-817, https://doi.org/10.5194/hess-22-789-2018, URL https://www.hydrol-earth-syst-sci.net/22/789/2018/, 2018.

- Lian, X., Piao, S. L., Huntingford, C., Li, Y., Zeng, Z. Z., Wang, X. H., Ciais, P., McVicar, T. R., Peng, S. S., Ottlé, C., Yang, H., Yang, Y., Zhang, Y., and Wang, T.: Partitioning global land evapotranspiration using CMIP5 models constrained by observations, Nature Climate Change, 8, 640–646, https://doi.org/10.1038/ s41558-018-0207-9, URL http://www.nature.com/articles/s41558-018-0207-9, 2018.
- Lian, X., Piao, S. L., Li, L. Z. X., Li, Y., Huntingford, C., Ciais, P., Cescatti, A., Janssens, I. A., Peñuelas, J., Buermann, W., Chen, A. P., Li, X. Y., Myneni, R. B., Wang, X. H., Wang, Y. L., Yang, Y. T., Zeng, Z. Z., Zhang, Y. Q., and McVicar, T. R.: Summer soil drying exacerbated by earlier spring greening of northern vegetation, Science Advances, 6, https://doi.org/10.1126/sciadv.aax0255, 2020.
- Liu, X. C., Liu, W. F., Yang, H., Tang, Q. H., Flörke, M., Masaki, Y., Müller Schmied, H., Ostberg, S., Pokhrel, Y., Satoh, Y., and Wada, Y.: Multimodel assessments of human and climate impacts on mean annual streamflow in China, Hydrology and Earth System Sciences, 23, 1245–1261, https://doi.org/10.5194/hess-23-1245-2019, URL https://www.hydrol-earth-syst-sci.net/23/1245/2019/, 2019.
- Nash, J. and Sutcliffe, J.: River flow forecasting through conceptual models part I A discussion of principles, Journal of Hydrology, 10, 282-290, https://doi.org/10. 1016/0022-1694(70)90255-6, URL https://linkinghub.elsevier.com/retrieve/ pii/0022169470902556, 1970.
- Ngo-Duc, T., Laval, K., Ramillien, G., Polcher, J., and Cazenave, A.: Validation of the land water storage simulated by Organising Carbon and Hydrology in Dynamic Ecosystems (ORCHIDEE) with Gravity Recovery and Climate Experiment (GRACE) data, Water Resources Research, 43, 1–8, https://doi.org/10.1029/2006WR004941, 2007.
- Pan, S. F., Pan, N. Q., Tian, H. Q., Friedlingstein, P., Sitch, S., Shi, H., Arora, V. K., Haverd, V., Jain, A. K., Kato, E., Lienert, S., Lombardozzi, D., Nabel, J. E. M. S., Ottlé, C., Poulter, B., Zaehle, S., and Running, S. W.: Evaluation of global terrestrial evapotranspiration using state-of-the-art approaches in remote sensing, machine learning and land surface modeling, Hydrology and Earth System Sciences, 24, 1485–1509, https://doi.org/10.5194/hess-24-1485-2020, URL https://hess.copernicus.org/articles/24/1485/2020/, 2020.
- Scanlon, B. R., Zhang, Z. Z., Save, H., Sun, A. Y., Müller Schmied, H., van Beek, L. P. H., Wiese, D. N., Wada, Y., Long, D., Reedy, R. C., Longuevergne, L., Döll, P., and Bierkens, M. F. P.: Global models underestimate large decadal declining and rising water storage trends relative to GRACE satellite data, Proceedings of the National Academy of Sciences, 115, E1080–E1089, https://doi.org/10.1073/pnas.1704665115, URL http://www.pnas.org/lookup/doi/10.1073/pnas.1704665115, 2018.

- Tang, Q. H., Oki, T., Kanae, S., and Hu, H. P.: Hydrological Cycles Change in the Yellow River Basin during the Last Half of the Twentieth Century, Journal of Climate, 21, 1790–1806, https://doi.org/10.1175/2007JCLI1854.1, URL http: //journals.ametsoc.org/doi/abs/10.1175/2007JCLI1854.1, 2008.
- Wada, Y., Wisser, D., and Bierkens, M. F. P.: Global modeling of withdrawal, allocation and consumptive use of surface water and groundwater resources, Earth System Dynamics, 5, 15–40, https://doi.org/10.5194/esd-5-15-2014, URL http://www. earth-syst-dynam.net/5/15/2014/, 2014.
- Wada, Y., de Graaf, I. E. M., and van Beek, L. P. H.: High-resolution modeling of human and climate impacts on global water resources, Journal of Advances in Modeling Earth Systems, 8, 735–763, https://doi.org/10.1002/2015MS000618, URL http://doi.wiley.com/10.1002/2015MS000618, 2016.
- Wang, S. S., Mo, X. G., Liu, S. X., Lin, Z. H., and Hu, S.: Validation and trend analysis of ECV soil moisture data on cropland in North China Plain during 1981-2010, International Journal of Applied Earth Observation and Geoinformation, 48, 110-121, https://doi.org/10.1016/j.jag.2015.10.010, URL http: //www.sciencedirect.com/science/article/pii/S0303243415300441http: //linkinghub.elsevier.com/retrieve/pii/S0303243415300441, 2016.
- Wang, X. H.: Impacts of environmental change on rice ecosystems in China: development, optimization and application of ORCHIDEE-CROP model, Ph.D. thesis, Peking University, 2016.
- Wang, X. H., Ciais, P., Li, L., Ruget, F., Vuichard, N., Viovy, N., Zhou, F., Chang, J. F., Wu, X. C., Zhao, H. F., and Piao, S. L.: Management outweighs climate change on affecting length of rice growing period for early rice and single rice in China during 1991–2012, Agricultural and Forest Meteorology, 233, 1–11, https://doi.org/10. 1016/j.agrformet.2016.10.016, URL http://linkinghub.elsevier.com/retrieve/ pii/S0168192316304087, 2017.
- Wu, X. C., Vuichard, N., Ciais, P., Viovy, N., de Noblet-Ducoudré, N., Wang, X. H., Magliulo, V., Wattenbach, M., Vitale, L., Di Tommasi, P., Moors, E. J., Jans, W., Elbers, J., Ceschia, E., Tallec, T., Bernhofer, C., Grünwald, T., Moureaux, C., Manise, T., Ligne, A., Cellier, P., Loubet, B., Larmanou, E., and Ripoche, D.: ORCHIDEE-CROP (v0), a new process-based agro-land surface model: model description and evaluation over Europe, Geoscientific Model Development, 9, 857–873, https://doi.org/10. 5194/gmd-9-857-2016, URL http://www.geosci-model-dev-discuss.net/8/4653/ 2015/http://www.geosci-model-dev.net/9/857/2016/, 2016.
- Xi, Y., Peng, S. S., Ciais, P., Guimberteau, M., Li, Y., Piao, S. L., Wang, X. H., Polcher, J., Yu, J. S., Zhang, X. Z., Zhou, F., Bo, Y., Ottle, C., and Yin, Z.: Contributions of Climate Change, CO 2, Land-Use Change, and Human Activities to Changes

in River Flow across 10 Chinese Basins, Journal of Hydrometeorology, 19, 1899–1914, https://doi.org/10.1175/JHM-D-18-0005.1, URL http://journals.ametsoc.org/doi/10.1175/JHM-D-18-0005.1, 2018.

- Yin, Z., Ottlé, C., Ciais, P., Guimberteau, M., Wang, X. H., Zhu, D., Maignan, F., Peng, S. S., Piao, S. L., Polcher, J., Zhou, F., and Kim, H.: Evaluation of ORCHIDEE-MICT-simulated soil moisture over China and impacts of different atmospheric forcing data, Hydrology and Earth System Sciences, 22, 5463-5484, https://doi.org/10.5194/hess-22-5463-2018, URL https://www.hydrol-earth-syst-sci-discuss.net/hess-2017-699/https: //www.hydrol-earth-syst-sci.net/22/5463/2018/, 2018.
- Yin, Z., Wang, X. H., Ottlé, C., Zhou, F., Guimberteau, M., Polcher, J., Peng, S. S., Piao, S. L., Li, L., Bo, Y., Chen, X. L., Zhou, X. D., Kim, H., and Ciais, P.: Improvement of the Irrigation Scheme in the ORCHIDEE Land Surface Model and Impacts of Irrigation on Regional Water Budgets Over China, Journal of Advances in Modeling Earth Systems, 12, 1–20, https://doi.org/10.1029/2019MS001770, URL https://onlinelibrary.wiley.com/doi/abs/10.1029/2019MS001770, 2020.
- Zhu, Z. C., Piao, S. L., Myneni, R. B., Huang, M. T., Zeng, Z. Z., Canadell, J. G., Ciais, P., Sitch, S., Friedlingstein, P., Arneth, A., Cao, C. X., Cheng, L., Kato, E., Koven, C., Li, Y., Lian, X., Liu, Y. W., Liu, R., Mao, J. F., Pan, Y. Z., Peng, S. S., Peñuelas, J., Poulter, B., Pugh, T. A. M., Stocker, B. D., Viovy, N., Wang, X. H., Wang, Y. P., Xiao, Z. Q., Yang, H., Zaehle, S., and Zeng, N.: Greening of the Earth and its drivers, Nature Climate Change, 6, 791–795, https://doi.org/10.1038/nclimate3004, URL http://www.nature.com/doifinder/10.1038/nclimate3004, 2016.

**Reply to Referee #2**

Z. Yin on behalf of all co-authors

1 "The study 'Irrigation, damming, and streamflow fluctuations of the Yellow River' by Yin et al. provides an overview of the water budget in the Yellow River basin, by considering irrigation and dam regulations. In this study, the authors developed a simple dam model coupled with ORCHIDEE to represent the major flow regulations in the river basin. The topic fits the scope of HESS, However, as a scientific manuscript, a clearly defined science question is missing in this study. What is your major contribution to the hydrology community as the concept of modeling dam regulation is not new?" A: Thank you very much for your comments. There are two objectives of this study. First, with newly developed crop and irrigation module, the land surface model OR-CHIDEE must be evaluated whether it is able to simulate the discharge of complex rivers with a generic parameterization and to explain the mismatch of simulated discharge of the Yellow River in our previous study (Xi et al., 2018). Moreover, the dam operation model should be evaluated before integrated into ORCHIDEE.

Second, we aim to quantify the impacts of irrigation and dam operations on the monthly discharge fluctuations of the Yellow River, which is not well demonstrated in previous studies. In the revised manuscript (Page 1, Line 3–6) we underlined: "This study aims to 1) demonstrate whether the global land surface model ORCHIDEE is able to simulate the streamflows of complex rivers with human activities using a generic parameterization, and 2) quantify the respective roles of irrigation and artificial reservoirs in monthly streamflow fluctuations of the Yellow River from 1982 to 2014 by using ORCHIDEE with a newly developed irrigation module, and an offline dam operation model." And in the introduction (Page 5, Line 100–101):"1) demonstrate whether ORCHIDEE and the dam model, with generic parameterizations, are able to reproduce streamflow fluctuations of the YR with human perturbations;..." In comparison to previous studies, there are several advantages in our work. Details are discussed in our reply to comment 1 of Referee #1.

2 "Page 1, line 5, line 10: new  $\rightarrow$  newly" A: Corrected.

3 "Page 4, lines 7-8: Although it's true that many dam model algorithms in recent GHMs and LSMs are inherited from Hanasaki et al. (2006), it is worth mentioning there are other types of dam/reservoir models such as agent-based models (e.g. Riverwave), or basin-specific models (e.g. USBR

**Colorado River Simulation System)."**

A: Thanks. We've added them in the short review of dam model development (Page 4, Line 58–61) as: "Although there are a set of dam models developed from different perspectives, such as agent-based model River Wave (Humphries et al., 2014) and basin-specific model Colorado River Simulation System (Bureau of Reclamation, 2012), the dam module in many global hydrological studies are based on the work of Hanasaki et al. (2006), which simulates dam operations based on different..."

**4 "Page 4, line 23: Remove 'real' before observations. Are there 'unreal' observations?"**

A: Sorry for the confusion. It has been removed.

5 "Page 4, lines 29-30: I'm not convinced that the new dam model 'does not require any prior information from observation'. In my opinion, observed information include the data or parameters measured/collected from the real world. In this case, the location, storage capacity, geometry of the dam and reservoir, etc. They are all 'observations'. So, I feel this sentence (and the one in the abstract) is a bit overselling the model and needs to be further clarified."

A: True. The dam model does require information like regulation capacity, location, and the year when regulation started. This part has been removed in the revision.

**6 "Section 2.1.1: Could you add some more background about ORCHIDEE before introducing ORCHIDEE-CROP? What's the relationship between these two? Is ORCHIDEE-CROP an offline crop model taking ORCHIDEE output as input, or it's an updated ORCHIDEE with an online crop model, or it's a regional model only focuses on China?"**

A: ORCHIDEE-CROP is a special branch of ORCHIDEE with an online crop model, which will be merged with the trunk version after extensive evaluation. It has been applied widely in current research. To avoid this confusion, we removed ORCHIDEE-CROP in the revision. A short introduction of ORCHIDEE and this special version has been added in the revision (Page 5, Line 109–117) as: "ORCHIDEE is a physical processbased land surface model that integrates hydrological cycle, surface energy balances, carbon cycle, and vegetation dynamics by two main modules. The SECHIBA (surfacevegetation-atmosphere transfer scheme) module simulates the dynamics of water cycle. energy fluxes, and photosynthesis at half-hourly time interval, which are used by the STOMATE (Saclay Toulouse Orsay Model for the Analysis of Terrestrial Ecosystems) to estimate vegetation and soil carbon cycle at daily time step. The ORCHIDEE used in this study is a special version with newly developed crop and irrigation module (Wang et al., 2017; Wu et al., 2016; Yin et al., 2020). The novel crop module includes specific parameterizations for three main staple crops: wheat, maize, and rice, which are calibrated over China by observations (Wang, 2016; Wang et al., 2017). It is able to simulate crop carbon allocation, different phenological stages as well as related managements (e.g., planting date, rotation, multi-cropping, irrigation, etc)."

**7 "Section 2.1.2: This scheme concept is quite similar to Voisin et al. (2013). Considering citing the work."**

A: Thanks. It has been cited in the introduction of the dam model framework (Page 6, Line 138–139) as: "Firstly, similar to Voisin et al. (2013), multi-year averaged monthly discharge ( $Q_s$ ) is calculated based on simulations..."

8 "Section 2.1.2: Essentially the dam model is trying to flatten the hydrograph. Any support from the observation that all dams follow this generic rule? I understand sometimes it's hard to obtain the actual operation rules from the dam operators, but given this is a basin scale analysis (not global), some level of 'fact-checking' needs to be included to reflect the local reality." A: The functions of main artificial reservoirs in the YRB has been collected from the Yellow River Conservancy Commission of the Ministry of Water Resources (http://www. yrcc.gov.cn/hhyl/sngc/), and has been added in Table 1 in the revised manuscript. The information confirms that flood control ('C' in Table 1), irrigation ('I'), and water supply ('W') are primary targets of these reservoirs, which, in principle, would flatten the hydrograph (seems impossible to release water for water supply and irrigation during flooding season, or reduce the discharge during the dry season).

9 "Page 8, line 22: Since NI and IR are major simulation experiments performed in this study, it is necessary to include more descriptions about the irrigation scheme in Section 2.1.1. For example, how does the irrigation demand be evaluated, at what time step? How does the irrigation water be applied, at what time step? I'm assuming different PFTs are associated with different irrigation methods (e.g. drip, sprinkler, or flood)? How does the return flow be treated in the model? How does the groundwater be represented in the model? If no groundwater pumping is represented in the model, the level of uncertainty needs to be evaluated and discussed for the study basin."

A: The irrigation demand is checked every half an hour. If water stress excesses predefined threshold, irrigation will be triggered. Due to lack of information about irrigation techniques for specific crops, only surface irrigation is applied. If irrigated rate is larger than the infiltration rate, surface runoff will occur, which however is almost forbidden by constraining the irrigation rate. To give a precisely estimation of irrigation consumption, the deep drainage of crop soil columns is turned off. Therefore, the irrigated water can only be used for evapotranspiration. Note that soil water in natural vegetation soil columns still can be lost by deep drainage, which forms the slow reservoir (shallow ground water) that can be withdrawn for irrigation as well. The fossil ground water pumping is not taken into account in our model. Firstly, the interactive mechanisms between shallow and fossil ground water is now well known (Scanlon et al., 2018). Secondly, there is rare data about the accessibility of deep fossil ground water. Nevertheless, in our previous study (Yin et al., 2020), by using ORCHIDEE-estimated irrigation water withdrawal and a proportion of surface water withdrawal versus ground water storage in the YRB (simulated trend is  $-5.4 \text{ mm.yr}^{-1}$ ; GRACE based trend is  $-5.36 \text{ mm.yr}^{-1}$ ).

We've improved the introductions of the irrigation module in Section 2.1.1 (Page 6, Line 123–127) as: "The water resources in ORCHIDEE account for three water reservoirs: 1) the stream reservoir indicates streamflows; 2) the fast reservoir indicates surface runoff; and 3) the slow reservoir indicates total deep drainage, the order of which indicates the priorities of water reservoirs considered for irrigation. As long-distance water transfer is not taken into account, streams only supply water to the crops growing in the grid-cell they cross, according to the river routing scheme of the ORCHIDEE model (Ngo-Duc et al., 2007)." and the simulation protocol in Section 2.4 (Page 9, Line 216–221) as: "In IR, only surface irrigation is considered in this study (irrigated water is applied on the cropland surface without interception by canopies), which only works during the crop growth period. The soil water stress, a function of profiles of soil moisture and crop root density (up to 2 m depth, (Yin et al., 2020)), is checked every half an hour. When it is less than a target threshold (=1), irrigation will be triggered with amount equal to the deficit of saturated and current soil moisture. To precisely estimate irrigation water consumption (direct water loss from the surface water pool excluding return flow), the deep drainage of the three crop soil columns is turned off in the IR simulation."

10 "Page 10, line 5: I don't understand why  $ET_{NI}$  and  $ET_{IR}$  had no significant differences as I can see the discharge had significant decreases at some gauges (Figure 3). I assume the reduced Q is due to the irrigation water withdrawal, and then become additional ET through the irrigation, or it's not the case here?"

A: Here we compared the magnitudes of simulated ET and observed (or satellite-based) ET, the differences between which is not significant (differences are smaller than the variation of observed ET among different products). In fact, simulated ET coincides well with the observations (Table S1). True. The ETIR is always higher than ETNI due to the irrigation withdrawal, which also results in  $Q_{\rm IR} < Q_{\rm NI}$ .

**11 "Page 10, line 9: In this equation, $A_i$ is the total drainage area between two gauges. Will it make more sense to use irrigated area instead of total area? This way you can compare the relative level of irrigation for different sub-regions?"**

A: Thanks for your suggestion. The equation here corresponds to the Equation 8. Here we provided sub-section-based water balance diagnosis. Although it is a good idea to show irrigation intensity (by changing  $A_i$  to irrigated area), we should consider the water balances in sub-sections, where precipitation and evapotranspiration – that are not only occur on irrigated cropland – are taken into account as well. The spatial distribution of irrigation intensity has been illustrated in our previous study (Yin et al., 2020).

**12 "Page 11, line 16: There are many negative spikes in $\hat{Q}_{IR}$ time series in Figure 5. This is unacceptable. I don't think your model is doing the right thing."**

A: Many thanks for your comment which allows us to find and correct an issue in our dam modelling. Indeed, the water recharge of reservoirs was not constrained by inflows

and that explains the negative spikes in  $\hat{Q}_{IR}$  time series. In the revision, we corrected corresponding equations (Eq. 6) and re-performed the simulations and results.

13 "Figure7: Given it's a regional study, I'm expecting better results than this, especially when you mentioned some previous study reached NSE around 0.9 for natural flow in the very same basin. Theoretically speaking, the inclusion of irrigation and dam regulation would improve the performance, not the opposite. I think more discussion about this issue is required. Also, how confident are you about the numbers in the conclusion?"

A: The inclusion of irrigation and dam regulation would dramatically reduce the RMSE, which has been shown in our result (MSE=RMSE2, Fig. 7a). However, it probably will not lead to a higher NSE of regulated discharge than NSE of naturalized discharge. Here is a simple proof.

Assuming that  $N_i$  is the time series of natural discharge and  $\Delta W_i$  is water storage change of a reservoir. Thus, the regulated discharge  $R_i$  can be calculated as:

$$R_i = N_i - \Delta W_i,$$

$$r_i = n_i - \Delta w_i.$$
(1)

Where i is month index. Capital letters indicate observed variables; while lower case letters indicate simulated variables. Then the NSE of regulated discharge (NSE1) can be calculated as:

$$NSE_{1} = 1 - \frac{\sum_{i=1}^{M} (R_{i} - r_{i})^{2}}{\sum_{i=1}^{M} (R_{i} - \bar{R})^{2}}$$

$$= 1 - \frac{\sum_{i=1}^{M} [(N_{i} - \Delta W_{i}) - (n_{i} - \Delta w_{i})]^{2}}{\sum_{i=1}^{M} (R_{i} - \bar{R})^{2}},$$
(2)

where M is the length of the time series. Let's assume that the model can give a perfect simulation of water storage change of reservoir. Thus  $\Delta w_i = \Delta W_i$  and NSE1 is,

NSE1 = 1 -
$$\frac{\sum_{i=1}^{M} (N_i - n_i)^2}{\sum_{i=1}^{M} (R_i - \bar{R})^2}$$
. (3)

Note that the NSE of natural discharge  $(NSE_2)$  is,

NSE2 = 1 -
$$\frac{\sum_{i=1}^{M} (N_i - n_i)^2}{\sum_{i=1}^{M} (N_i - \bar{N})^2}$$
. (4)

The difference between  $NSE_1$  and  $NSE_2$  is the variation of regulated and natural discharge. As assuming that dam operations always reduce the variation of discharge, the variation of  $N_i$  is smaller than  $R_i$ . Consequently, NSE2 is always less than NSE1. In summary, if reservoirs reduce the variation of river discharge, a model even with a perfect dam module will always provide a smaller NSE (with regulated discharge as reference) than that of the model without functions of dam operations (with natural discharge as reference)! The conclusion is that it is not comparable of model (study) performances with different references and that it is not adequate to evaluate dam parameterizations. This proof has been added in the online supplement. And in Sect. 4 (Page 14, Line 401 – Page 15, Line 405) we discussed: "These NSE decreases were interpreted due to the complexity of the YRB under the impacts of human activities and climate variation. However, the NSE of natural discharges is incomparable to the NSE of regulated discharges. Even if the model can perfectly simulate the reservoir operations, the NSE of natural discharges is certainly larger than that of regulated discharges from the same model, if you accept the assumption that reservoir operations reduce the variation of river s